

# Weather persistence on sub-seasonal to seasonal timescales: a methodological review

Alexandre Tuel[1] and Olivia Martius[1,2]

[1]Institute of Geography and Oeschger Centre for Climate Change Research, University of Bern, Switzerland
[2]Mobiliar Lab for Natural Risks, University of Bern, Switzerland

**Correspondence:** Alexandre Tuel (alexandre.tuel@giub.unibe.ch)

**Abstract.** Persistence is an important concept in meteorology. It refers to surface weather or the atmospheric circulation either remaining in approximately the same state (stationarity) or repeatedly occupying the same state (recurrence) over some prolonged period of time. Persistence can be found at many different timescales; however, the sub-seasonal to seasonal (S2S) timescale is especially relevant in terms of impacts and atmospheric predictability. For these reasons, S2S persistence has
been attracting increasing attention by the scientific community. The dynamics responsible for persistence and their potential evolution under climate change are a notable focus of active research. However, one important challenge facing the community is how to define persistence, from both a qualitative and quantitative perspective. Despite a general agreement on the concept, many different definitions and perspectives have been proposed over the years, among which it is not always easy to find one's way. The purpose of this review is to present and discuss existing concepts of weather persistence, associated methodologies and physical interpretations. In particular, we call attention to the fact that persistence can be defined as a global or as a local
property of a system, with important implications in terms of methods but also impacts. We also highlight the importance of timescale and similarity metric selection, and illustrate some of the concepts using the example of summertime atmospheric circulation over Western Europe.

## 1   Introduction

Surface weather persistence at sub-seasonal to seasonal (S2S) timescales can have severe impacts on human and natural systems. Long-lasting dry conditions, for instance, can lead to droughts and wildfires and can affect agriculture and energy production. Long-lasting wet spells may cause severe flooding and crop loss. Persistent surface weather can result from quasi-stationary, long-lived atmospheric circulation conditions or from recurrent, shorter-lived circulation features. Recurrence refers to the repeated occurrence of similar large-scale circulation types or weather systems within some time interval, usually with brief
interruptions. Many recent high-impact weather and climate events were linked to persistent quasi-stationary or recurrent weather conditions. For example, the Western European floods in July 2021 occurred at the end of an extreme wet spell in Western Europe that resulted from repeated atmospheric blocks and Rossby wave breaking episodes (Tuel et al., 2022b). Other examples include the floods in the UK during winter 2013-2014 and in Queensland (Australia) in February-April 2022 caused by sequences of cyclones (Huntingford et al., 2014) (Wikipedia, 2022; Floodlist, 2022). Catastrophic flooding in Pakistan



in summer 2022 resulted from persistent and particularly heavy monsoon rains (Mallapaty, 2022). Intense heatwaves and associated atmospheric circulations also tend to be persistent (Lorenz et al., 2010), as in Western Europe in 2003 (Black et al., 2004; Trigo, 2005), Western Russia in 2010 (Drouard and Woollings, 2018; Di Capua et al., 2021) or China (WMO, 2022) and India (Bloomberg, 2022) in 2022.

S2S weather persistence offers the potential for improved predictability at the S2S timescale (Franzke et al., 2011), which is
highly relevant for risk preparedness, and is attracting increased attention from the research community (e.g., Vitart et al., 2017; Meehl et al., 2021; Domeisen et al., 2022). However, a persistent state is not necessarily well predictable, and persistent states with low predictability can cause large errors in sub-seasonal weather forecasts (Quandt et al., 2017).

Persistence is also an important aspect of climate model evaluation and climate projections. Whether global climate models are able to correctly simulate persistence is key to the robustness of long-term projections, especially of high-impact weather
events – all the more so as climate projections suggest enhanced persistence (Li and Thompson, 2021; Hoffmann et al., 2021; Tuel and Martius, 2021a) in the future.

Characterising weather persistence is therefore key to our understanding of the atmospheric circulation and its predictability, and the associated hazards. Previous studies have focused on weather persistence from varied perspectives. Some assessed the persistence of specific weather systems or features, like atmospheric blocking (e.g. Liu, 1994), Rossby waves (Röthlisberger
et al., 2019), or teleconnection patterns (Barnes and Hartmann, 2010). Others analysed specific episodes of particularly persistent weather conditions (e.g. Black et al., 2004; Di Capua et al., 2021; Tuel et al., 2022b), while others yet characterised the overall tendency of the atmospheric circulation and surface weather to exhibit persistence (e.g. MacDonald, 1992; Li and Thompson, 2021; Hoffmann et al., 2021). However, while previous studies generally agree on what persistence means conceptually, past work on this topic has involved many different definitions, often causing confusion and leading to different interpretations of
persistence. Many case studies have also described observed situations as persistent based on subjective analyses rather than quantitative metrics. A further source of confusion is that "recurrence" is used in the literature to refer not only to successive occurrences of the same weather pattern at close intervals – what we will focus on in this review – but also to the states of the atmosphere with the highest probability of occurrence (Michelangeli et al., 1995).

It is difficult to give a unique definition of weather persistence. Besides, it may not even be desirable as different interpretations
are possible and useful, depending on the system and timescale of analysis, and on the motivations and goals of the study. Our goal here is therefore to review existing concepts of weather persistence, associated methodologies and physical interpretations. We present and structure a wide variety of approaches, definitions, techniques and metrics that have been used to analyse these concepts and that allow answering one or several of the following questions:

- Is there persistence in the data?

– At which timescale(s) does the persistence occur?

- In which specific periods does persistence occur?

- What are the persistent locations in the phase space?



While persistence occurs at many different timescales, we specifically focus on the S2S timescale – often the most impact-relevant, and certainly important for predictability – but most methods and arguments apply in principle also to longer timescales.
We begin in Section 2 by introducing the two aspects of persistence: stationarity and recurrence. Section 3 then discusses several different perspectives on persistence to consider when choosing an analysis methodology. Finally, we present a detailed list of methods to detect or quantify persistence in Sections 4 and 5. We illustrate most of the methods with examples taken from the literature or with our own analyses of summertime atmospheric circulation over Europe (details about the data we use are given in the appendix). We keep the interpretation of the results to a minimum, the point being to illustrate the methods and not to
analyse European summer circulation persistence in detail.

## 2   Persistence: stationarity or recurrence?

Persistence in a dynamical system (climate variable, atmospheric circulation field, etc.) arises from the repeated occurrence of the same value(s) or pattern(s) over a period of time. Successive occurrences can follow each other continuously – a situation we refer to as "stationary" – or in an interrupted sequence – in which case we speak of "recurrence". Persistence therefore comes in
two flavours, stationarity and recurrence, which we illustrate with the example of extreme warm conditions in a region north of the Black Sea in Figure 1. Here, extreme warmth is defined as a daily-mean temperature exceeding 1.5 standard deviations above its annual cycle. In summer 2010, the region experienced persistent (stationary) extreme warmth, with temperatures remaining continuously above the threshold for 41 days straight (Figure 1-a). By contrast, in spring 2019, extreme warm temperatures occurred frequently throughout May, but the three extreme warm episodes within one month were separated by at least 5 days
with near-average temperatures (Figure 1-b). The corresponding evolution of the atmospheric circulation can be conceptually described as a slow-moving trajectory for stationary conditions (Figure 1-b) and a more rapidly-evolving trajectory that revisits the same point repetitively (Figure 1-d).

The literature often uses "persistence" as a synonym for "stationarity", (e.g., Dole and Gordon, 1983; Barnston and Livezey, 1987; Franzke et al., 2011; Di Lorenzo and Mantua, 2016; Fereday, 2017; Liu et al., 2018; Du et al., 2019; Francis et al., 2020;
Hoffmann et al., 2021; Li and Thompson, 2021), i.e., persistence is associated with long-lived flow anomalies and little change in atmospheric circulation and surface weather. Recurrence has by contrast attracted less scientific attention. Most studies on recurrence have focused on extreme/impactful events (e.g., Mailier et al., 2006; Barton et al., 2016; Dacre and Pinto, 2020; Tuel and Martius, 2022a) and Rossby waves (e.g., Röthlisberger et al., 2019; Ali et al., 2021). Yet, from a physical perspective, stationarity and recurrence are intimately related. Recurring weather systems typically result from stationary favourable large-
scale conditions, like sea-surface temperature anomalies or the location of extratropical jets (e.g., Tuel and Martius, 2022b). Temporal dependence, or memory, in a system can thus translate in practice into both stationarity and recurrence (Franzke, 2013). Additionally, from the impacts perspective, it makes sense to look at stationarity and recurrence together, since both can cause prolonged, impactful surface weather. Recurrent Rossby waves, for instance, modulate the persistence of surface temperature and precipitation anomalies (Röthlisberger et al., 2019; Ali et al., 2021) while long-lived SST anomaly patterns
like ENSO can remotely trigger recurrent extreme weather (Gershunov and Barnett, 1998).



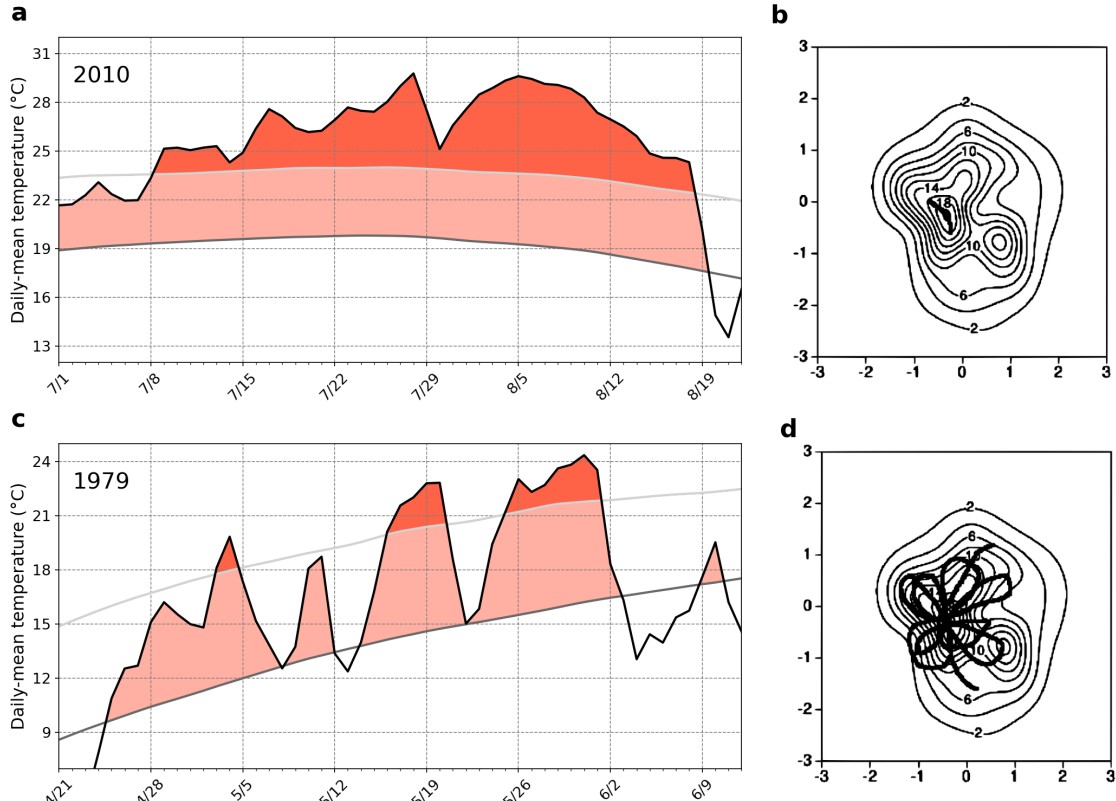

**Figure 1. Illustrating stationarity and recurrence on S2S timescales.** (a,c) Example of daily mean 2-meter temperature series averaged over the 33°-43°E/50°-57°N region illustrating (a) stationarity (summer 2010) and (c) recurrence (spring 1979) in extreme warm temperatures on S2S timescales (black: observations; gray: mean annual cycle; light gray: +1.5 standard deviation from the mean). Data is from the ERA5 reanalysis (Hersbach et al., 2020). (b,d) Idealised system trajectories (thick black lines) in the phase space corresponding to stationarity (b) and recurrence (d). The background PDF is shown in light contours. Panels (b,d) reproduced with permission from Hannachi et al. (2017).

Note that "recurrence", as we define it here, is sometimes referred to as "temporal" or "serial clustering" (Franzke, 2013), for instance in the case of recurrent cyclones (Mailier et al., 2006) or heavy precipitation events (Barton et al., 2016). Note also that "recurrence" is also frequently used in meteorology to refer to preferred patterns that repeatedly occur in a time series, but not necessarily at close intervals (e.g., Vigaud et al., 2018; Kornhuber et al., 2019; Son et al., 2021) – in the case of
atmospheric circulation patterns, one speaks of circulation or weather "regimes" (Michelangeli et al., 1995). This is different from our definition, in which recurrence specifically relates to the repeated occurrence of the same patterns over S2S timescales. Such patterns may be rare in the full dataset and would therefore not be considered as "recurrent" in the regime approach, but can be highly relevant from an impacts perspective. Hannachi et al. (2017) gave a comprehensive overview of the weather regime approach and associated methodologies. We also discuss the relevance of the weather regime perspective for stationarity
analysis in section 4.2.1.





Because stationarity and recurrence are two faces of the same coin, distinguishing one from the other may not be evident nor necessarily relevant.

First, the distinction often depends on the variable of interest. Recurrent weather systems can indeed result in stationary surface weather anomalies, and vice-versa. For example, the long-lived (stationary) heatwaves of the 2010 and 2021 summers in Western
Russia (Fig. 1-a) and the Baltic were linked to recurrent atmospheric blocks (Drouard and Woollings, 2018; Tuel et al., 2022b). Likewise, prolonged droughts or wet spells may result from recurrent Rossby wave activity (Röthlisberger et al., 2019; Ali et al., 2021, 2022). Stationary surface warm and humid conditions can also trigger recurrent thunderstorm activity (Mohr et al., 2020). Conversely, recurrent extreme precipitation events (e.g., Barton et al., 2022) or extratropical cyclones (e.g., Dacre and Pinto, 2020) are often linked to stationary jet states that last for much longer than the lifetime of individual weather systems.

Second, the longer the timescale of analysis, the less obvious the difference between stationarity and recurrence becomes. Impacts for example often depend on anomalies of surface temperature or precipitation averaged or accumulated over several weeks to months (e.g., droughts). Weekly or monthly values may thus sometimes be preferred to daily ones, in which case synoptic-scale stationarity and recurrence would both result in large weekly or monthly anomalies that would result from simple "persistence".

Nevertheless, the distinction between stationarity and recurrence remains highly relevant for several reasons: from a methodological perspective (sections 4 and 5), but also for forecasting and process understanding at the synoptic timescale, as well as for some stakeholders, like insurers (for whom it matters whether impacts resulted from a single or multiple events). For process understanding, taking a weather systems perspective is often very relevant. In such a case, distinct weather systems (like cyclones) can in principle be separated from one another, and long-lived single systems be distinguished from multiple
short-lived systems occurring in close succession. It therefore matters whether persistence is driven by recurrence or stationarity. The distinction is also important to assess whether numerical models simulate persistence for the right physical reasons.

## 3 Considerations and methodological approaches for persistence analysis in weather and climate data

Before reviewing how persistence can be apprehended and quantified, let us begin with some basic notations and definitions. In the following, we denote by $(\mathbf{x}(t))_t \in \mathbb{R}^m$ the dynamical system under analysis.

$\mathbf{x}(t)$ could for example be the weather evolution in a region, the time evolution of one variable at one location, or the general circulation. $(\mathbf{x}(t))_t$ evolves within a phase space $\mathcal{S}$ that consists of the set of all possible *system states* $\{s_i\}_i$ that the system can occupy. To each time $t$ corresponds a single *system value*, $\mathbf{x}(t)$, and a single system state. The same system state can be attained at multiple different time steps. The series of successive system values constitutes the system *trajectory*. In practice, only a finite number of observations are available $(\mathbf{x}(t_k))_{1 \leq k \leq N}$ to characterize the persistence of a system.

Characterising a system as "persistent" can mean different things. Consequently, it is important to always specify the perspective that is taken to avoid confusion. First, there are different flavours to persistence (global, state or episodic persistence: section 3.1). Second, persistence can be studied from a Lagrangian or a Eulerian perspective (section 3.2). Finally, persistence is linked to a similarity metric (section 3.3) and a notion of timescale (section 3.4), for which different approaches are possible.





## 3.1 Global, state and episodic persistence

Persistence (whether stationarity or recurrence) is a broad concept that covers different kinds of behaviour in dynamical systems. We might say, for instance, that temperature is more persistent than precipitation because temperature evolves, on average, over longer timescales than precipitation. In this sense, persistence quantifies the system's inertia. However, if we qualify last summer's weather as particularly persistent, we mean something different: namely, that last summer's weather varied much less than what one could have reasonably expected in a normal summer. Here, persistence refers to some unusual behaviour of the

system over a particular period. And saying that zonal jets or atmospheric blocks are persistent means again something else: that these particular states of the circulation tend to be more long-lived or recurrent than other states.

This leads us to make the distinction between three types of persistence: global, state and episodic persistence.

1. *Global persistence* characterises the tendency to stationarity or recurrence across the whole trajectory of the dynamical system. We choose this term because persistence in this sense is a "global" property of the system: it is not restricted to

any particular system state or time period. Global persistence can sometimes be impact-relevant (e.g., trends in global persistence under climate change can be important for impacts). However, it does not make distinctions between the different system states. Consequently, global persistence is not suited to characterise persistent system states or specific time intervals with persistent system behaviour and is generally not the best-suited approach for risk assessment. Global persistence is, however, strongly related to intrinsic system predictability, and can thus yield important information for

numerical forecasting, including at the S2S timescale. Most global persistence methods focus on stationarity – like the autocorrelation coefficient (section 4.1.1) or the Hurst exponent (section 4.1.2) – but some exist for recurrence as well (section 5.6). Many studies look at global persistence: we may cite, for instance, MacDonald (1992); Pfleiderer and Coumou (2018); Pfleiderer et al. (2019) and Li and Thompson (2021) who analyse the persistence of temperature series and Hoffmann et al. (2021) who consider 10-day atmospheric flow persistence.

2. *State persistence* relates to the persistent behaviour of specific system states. Phase space persistence analysis consists in either identifying persistent system states – e.g. with the quasi-stationary (section 4.2.2) or the optimally persistent pattern (section 4.2.3) methods – or characterising the persistence of given states – with methods like residence times (section 4.2.5) for stationarity or Ripley's K for recurrence (section 5.3). State persistent is highly relevant from the impacts, the forecasting and the process understanding perspectives. Knowing which states are persistent, and to what extent, is useful

i) to make the link to surface impacts and ii) to know which weather patterns or sequences may be more predictable than others. It also helps to determine the physical processes that support or are responsible for the persistence. The state approach to persistence has been used to shed light on stationary states of the North Atlantic circulation and their predictability (Faranda et al., 2017b), to characterise the stationarity of continental-scale weather patterns (Francis et al., 2018, 2020) or drought (Ford and Labosier, 2014), or to quantify recurrence in extra-tropical cyclones (Mailier et al.,

2006) and extreme precipitation (Tuel and Martius, 2021a).





3. *Episodic persistence* is tied to specific time intervals (or "episodes") during which the system exhibits stationary or recurrent behaviour. It is in that sense a purely local property that characterises the anomalous behaviour of the system over a limited time period. Importantly, episodic persistence can occur simply by chance in any dynamical system, even in systems that exhibit no global persistence. Episodic persistence is therefore not necessarily relevant for predictability and process understanding. Still, persistent periods can always be analysed to look for potential drivers that discriminate pure "statistical flukes" from possibly predictable events. Episodic persistence is however well-suited for impacts analysis, because it can make a direct link between periods of persistent weather and impacts. Relevant methods include running window techniques (section 4.3) for stationarity and window counts (section 5.1) for recurrence. Hoffmann et al. (2021) investigated for example stationarity in 10-day sequences of atmospheric circulation, while Bevacqua et al. (2020) and Kopp et al. (2021) looked at sub-seasonal periods of with recurrent cyclones and extreme precipitation events.

While it helps to capture the various interpretations of persistence, this classification is not perfect, and there is some overlap between categories. In practice, the state or episodic perspectives can also be used to quantify global persistence (by averaging persistence metrics across systems states or time intervals) and global persistence metrics can be computed on subsets of the data to identify persistent periods. Some methods, like recurrence plots (section 5.6), can even deal with all three types of persistence.

The three types of persistence are also not independent from one another. Global persistence, for instance, can emerge from repeated occurrences in one or a handful of system states only, while the rest of the trajectory, if analysed separately, may not be qualified as persistent. There are also strong relationships between state and episodic persistence. The most common system states to occur during persistent periods are indeed likely to be persistent states. Symmetrically, persistent states, when they occur, are likely to be associated with persistent periods. Persistent states can thus be uncovered from the knowledge of persistent periods, for instance with pattern recognition or clustering algorithms applied to system values during persistent periods, or simply by averaging system values during persistent periods (e.g., Faranda et al., 2017b; Hoffmann et al., 2021).

However, state persistence only characterises the average behaviour of system states. Consequently, there is no one-to-one relationship between persistent states and persistent periods. Some system states can behave in a persistent way under certain conditions but not under others. The lifetime and travel speed of atmospheric blocks, for instance, is affected by land-atmosphere feedbacks or upstream latent heating (Steinfeld et al., 2020). Similarly, extratropical cyclones may occur in sequences but also as single events (Dacre and Pinto, 2020). Additionally, persistent periods may be associated with a variety of system states. While we expect persistent states to be the most frequent ones during persistent periods, some states which on average are not persistent can still at times by pure chance behave persistently.

Still, this classification is useful to illustrate the different methodological ways that weather persistence can be tackled, and we rely on it to structure the description of methods in sections 4 and 5.





## 3.2 Lagrangian and Eulerian perspectives

Weather persistence is most often analysed from a Eulerian perspective, i.e., persistence of the same weather pattern or conditions at a fixed location in space. By contrast, in the Lagrangian perspective the focus is on the persistence of a given weather pattern in time. In the Eulerian framework, the system $\mathbf{x}(t)$ represents a time series over a domain fixed with time, whereas in the Lagrangian one, $\mathbf{x}(t)$ follows individual weather patterns or the background atmospheric flow. For example, temperature anomalies can be tracked over time Kornhuber and Tamarin-Brodsky (e.g., 2021) or analysed at a fixed location Pfleiderer and Coumou (e.g., 2018) and Li and Thompson (2021). Similarly, weather systems such as blocking, cyclones or vortices (Bray and Cavallo, 2022) can be analysed at a fixed location or following the weather systems. For example, Økland and Lejenäs (1987) contrast the persistence of blocking at fixed longitudes against the persistence of individual blocking episodes. Another example is Kossin (2018), who characterise the persistence in tropical cyclones by their translation speed.

Both the Eulerian and the Lagrangian perspectives are relevant for impact and risk assessment. The former links persistence to impacts at a given location, and the latter highlights impacts along the trajectory of a weather pattern (system) during its lifetime. Indeed, the same weather system can produce hazardous weather over large areas, putting strain on the resources of insurance companies or governments. The two perspectives can be brought together by considering the translation speed and lifetime of the tracked weather systems. Systems with long lifetimes and low translation speed lead to both Eulerian and Lagrangian persistence. By contrast, long-lived systems that travel fast are persistent from a Lagrangian perspective only. Likewise, slow-moving but short-lived systems are not Lagrangian-persistent, but Eulerian persistence can still be detected if several such systems occur over the same area in close succession.

## 3.3 Quantifying similarity

Assessing persistence typically requires quantifying the self-similarity of system values $\mathbf{x}(t)$ with a metric (e.g., Wharton et al., 2008; Zerzucha and Walczak, 2012; Ali et al., 2020; Ontañón, 2020). Metric selection is an important step that should be done with care, because it conditions how persistence is quantified and some metrics are not suited to certain kinds of data (e.g., heavily skewed).There are two main classes of similarity metrics:

1. *categorical metrics*: with categorical metrics, system values are either similar (if they belong to the same category) or not (if they don't).

2. *continuous metrics*: continuous metrics measure the degree of similarity between two system values in a continuous way.

Categorical metrics focus on specific features of the system, like the presence of a given weather pattern or the occurrence of a specific event. They require the set of $(\mathbf{x}(t))_t$ values to be classified into distinct categories (usually from 2 to a few dozen). Categories can represent pre-defined system states of interest (e.g., a warm/cold anomaly, a given phase of a teleconnection mode, or the occurrence of a specific weather pattern, like a block) (e.g., Pinto et al., 2014; Drouard and Woollings, 2018; Pfleiderer and Coumou, 2018; Ali et al., 2021; Kopp et al., 2021), but can also be obtained objectively with dimension reduction/pattern recognition methods. Examples include Principal Component Analysis (or Empirical Orthogonal Functions,





EOF) (Fereday, 2017, e.g.,), Self-Organising Maps (SOM) (e.g., Francis et al., 2018; Weiland et al., 2021; Rousi et al., 2022b)

or clustering algorithms (probabilistic, hierarchical or non-hierarchical) (e.g., Demuzere et al., 2011; Hannachi et al., 2012; Grams et al., 2017). When $\mathbf{x}(t)$ represents 2D circulation data (like sea-level pressure or geopotential height), the ensemble of system states is often referred to as "weather regimes" (Michelangeli et al., 1995; Grams et al., 2017; Francis et al., 2018) (see section 4.2.1). In EOF analysis, the distance metric is the $L^2$ norm (Euclidean distance), but most clustering methods can work with any distance metric. The challenge with dimension reduction methods is choosing the number of categories to retain

(EOFs, clusters, SOM nodes, etc.). A high number can capture rare states of the system, but at the cost of making persistence more difficult to assess (since sequences when the system remains in the exact same category will become less frequent).

Continuous metrics measure the degree of similarity between any pair of system values in a continuous way. Common examples of continuous metrics include the Euclidean distance (e.g., Faranda et al., 2017b), pattern correlation(e.g., Mo and Ghil, 1987), or more complex similarity indices like the Teweles–Wobus score (e.g., Horton et al., 2017; Blanchet et al., 2018)

or the image structural similarity index (SSIM) (e.g., Hoffmann et al., 2021).

In comparison to categorical metrics, continuous metrics offer the advantage that they do not require specifying features/events of interest beforehand. They are also more flexible insofar as similarity can be quantified with respect to any system value as reference, and not just representative values for each category. Since they do not require simplifying the phase space, continuous metrics may also be able to pick up rare persistent patterns that are missed by dimension reduction methods that focus on the

most common patterns in a series. These advantages come at a cost: working with continuous metrics, especially complex ones, can be more computationally intensive, and sometimes more difficult to interpret physically.

### 3.4   Persistence timescales

Persistence is linked to a notion of timescale during which the system continuously remains in the same state (for stationarity) or occupies that same state repeatedly (for recurrence). There are three common ways to approach persistence timescales.

The first option is to choose a single, fixed timescale for analysis. This choice can be guided by impact and forecasting considerations, by physical knowledge of the underlying system, or by observations of persistent events (e.g., Huntingford et al., 2014; Lawrence et al., 2020; Overland and Wang, 2021; Rakovec et al., 2022; Rousi et al., 2022a). This is the most common approach to analyse episodic persistence, but it is also applicable to global persistence. Stationarity can be quantified by the average similarity between $n$ successive states (for continuous similarity metrics Kolstad et al. (see e.g., 2017); Hoffmann et al.

(see e.g., 2021)) or by the variety of system states during an $n$-step window (for categorical metrics Fereday (see e.g., 2017); Richardson et al. (see e.g., 2019)). Stationarity can also be inferred from extreme anomalies of circulation, temperature or precipitation during $n$-step windows. For instance, Gálfi et al. (2019) and Tuel and Martius (2023) identify persistent warm and cold spells by averaging temperature anomalies over 1-3 weeks.

Recurrence can similarly be assessed by calculating the number of times that a particular system state or event occurs during $n$-

step windows (e.g., Mailier et al., 2006; Pinto et al., 2014; Kopp et al., 2021; Tuel and Martius, 2022a). A fixed timescale can also be used as a threshold to separate stationary from non-stationary events, by requiring stationary events to last at least $n$ steps. For instance, Francis et al. (2018) and Francis et al. (2020) define persistent periods by requiring the circulation pattern in a region



to remain in the same state for at least four consecutive days. Mann et al. (2018) similarly define persistent resonant wave events as those lasting at least 10 days. Dole and Gordon (1983) also take this approach for the persistence of point-wise geopotential
anomalies. Note that this fixed timescale can also consist of a single time step. In this case, "persistence" characterises how much past system values determine future ones. Li and Thompson (2021), for instance, characterise persistence with the lag-1 autocorrelation coefficient, Röthlisberger and Martius (2019) look at 1-day transition probabilities between different system states, and Vautard (1990); Michelangeli et al. (1995) and Hannachi et al. (2017) calculate time derivatives of geopotential fields to identify quasi-stationary states.

The second option is to select an analysis method that explores a range of timescales and pinpoints the relevant persistence timescales, sometimes accompanied by some notion of statistical significance. This approach only works for global persistence. Autocorrelation analysis, for instance, detects the timescales at which the system exhibits significant lagged memory (section 4.1.1). Spectral analysis can likewise highlight important timescales of variability that can be linked to stationarity (section 4.1.2). For recurrence, methods like Ripley's K function indicate the timescales at which recurrence is statistically significant
(section 5.3).

Finally, the third option is to characterise persistence not by a single timescale, but by a distribution of timescales. To assess stationarity, one can typically work with the distribution of persistent event durations. Persistent events are periods during which the system verifies a persistence criterion: for stationarity successive system values must be similar; and for recurrence the same event must occur multiple times, each occurrence being separated by at most $n$ time steps from the previous one. For
recurrence, it is also possible to consider the distribution of inter-event times (e.g., Altmann and Kantz, 2005). The distribution of event lengths can be further summarised by considering the average or maximum event length (Faranda et al., 2017b; Rousi et al., 2022b), or by modeling it with an exponential or power-law distribution (section 4.2.5). This approach has been applied to numerous cases: heatwaves (Lorenz et al., 2010), droughts (Meng et al., 2017; Moon et al., 2018), wet spells (Ali et al., 2021), atmospheric blocks (Liu, 1994), circulation patterns (Huguenin et al., 2020), and mid-latitude cyclone clustering (Bevacqua
et al., 2020).

## 4 Stationarity

The diversity of perspectives on persistence translates into a wide range of methods, of which we give an overview in the following two sections dedicated respectively to stationarity and recurrence. Following the distinctions introduced in section 3, we separate methods that quantify global, state, and episodic persistence (though some methods can be used for more than
one). We also specify whether methods can only be used with a single timescale, or whether they quantify persistence across timescales. An overview of methods is shown in Table 1.

### 4.1 Global stationarity

We begin with several methods that quantify global stationarity in one-dimensional time series. They characterise stationarity in the series as a whole, but are generally unable to identify stationary states or periods. However, they often allow the user



to characterise the timescales of variability and persistence in the data, and are hence relevant for system predictability and process understanding.

### 4.1.1 Autocorrelation

Autocorrelation is a frequently used measure of stationarity in weather and climate science. If $X_t$ is a continuous, one-dimensional process of mean $\mu$ and variance $\sigma^2$, its Pearson autocorrelation coefficient at lag $k$ is defined as:

$$\rho(k) = \frac{\mathbb{E}\left[(X_t - \mu)(X_{t+k} - \mu)\right]}{\sigma^2} \tag{1}$$

Potential cycles and long-term trends should be removed from the data before analysis (Weiss and Weiss, 1999). In 1, $X_t$ can also be replaced by its rank, which yields the alternative Spearman autocorrelation, more robust to non-linear behaviour in the data. Several summary metrics for autocorrelation exist, like the autocorrelation timescale:

$$T_\rho^e = \min_{k>0} \{\rho(k) \le \exp(1)\} \tag{2}$$

the decorrelation time (Hannachi, 2021)

$$T_\rho^d = 1 + 2 \sum_{k=1}^{N} \rho(k) \tag{3}$$

where N is the number of timesteps (if $\rho$ is integrable), or the characteristic time $T_\rho^c$ (Trenberth, 1984) (Figure 2-a):

$$T_\rho^c = 1 + \sum_{k=1}^{N} \left(1 - \frac{k}{N}\right) \rho(k) \tag{4}$$

Both $T_\rho^d$ and $T_\rho^c$ roughly approximate the average time between successive independent values. The lag-1 autocorrelation

$\rho(1)$ can also be used as a stationarity metric (Li and Thompson, 2021), which is equivalent to fitting a linear regression model between successive system values: $\mathbf{x}(t+1) = \beta \mathbf{x}(t) + \epsilon(t)$, where $\epsilon(t)$ is a white-noise process. If $\mathbf{x}(t)$ is normalised, the regression slope $\beta$ is equal to $\rho(1)$. Note that more complex models are possible: linear models for continuous series can be extended by the large class of Autoregressive-Moving Average (ARMA) models.

Autocorrelation has many advantages: a simple definition, ease of interpretation ($\rho(k)$ is related to the linear regression

coefficient of $X_{t+k}$ against $X_t$), and high flexibility; it is already implemented in common programming languages, it requires no subjective threshold choice, and results can be easily reproduced. By varying the lag $k$, it can measure stationarity at all timescales. It also comes with a notion of statistical significance: given a confidence level, it is possible to say whether the obtained autocorrelation is significant (indicating a link at lag $k$) or not, pointing to the relevant stationarity timescales in the series. On the downside, autocorrelation only works for one-dimensional data and requires a large number of values as

input. Therefore it is best suited to measuring stationarity globally. One can compute autocorrelation on a subset of the data only (summer values or a specific time interval, for instance) if the number of data points is large enough, in which case autocorrelation may be used to characterise stationarity locally in time. Autocorrelation measures the strength of the connection



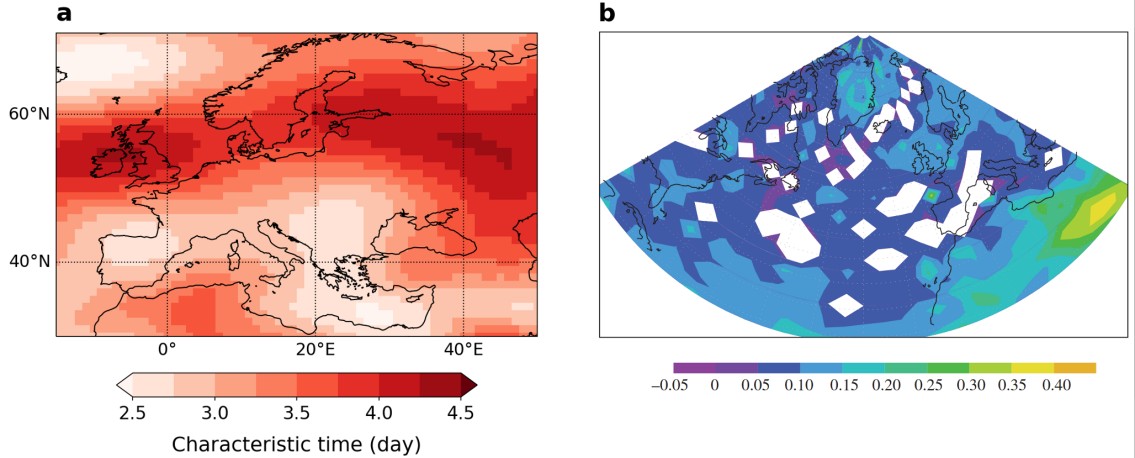

**Figure 2.** (a) Autocorrelation characteristic time ($T_\rho^c$; equation 4) of JJA daily rescaled Z500. (b) Long-range dependence parameter $d$ of unfiltered surface wind speeds. Only those values that are significant at the 5% level are displayed. Reproduced with permission from Franzke (2013).

between lagged system values, but not how far apart they might be in the phase space.

Example applications include Horel (1985a) and Barnston and Livezey (1987) who calculated lag autocorrelation on principal
component time series of monthly Northern Hemisphere geopotential fields. MacDonald (1992) used autocorrelation to detect stationarity in monthly temperature series, as did Weiss and Weiss (1999) to assess stationarity in ENSO. Weatherhead et al. (2010) characterised daily temperature stationarity in the Arctic based on lag-1 autocorrelation. Degenhardt and Ólafsson (2019) and Kolstad et al. (2015) calculated lag-1 autocorrelation to highlight intra-seasonal stationarity of monthly-mean temperatures in Iceland and Europe, respectively, and Li and Thompson (2021) applied autocorrelation to daily temperature series and found
it was strongly related to the average length of warm and cold episodes across the world. Kolstad et al. (2017) even used autocorrelation analysis in a causal discovery framework by regressing temperature values against previous ones and including potential covariates.

We show on Figure 2-a the characteristic time for daily rescaled 500 hPa geopotential height over Europe during summer. While the results say nothing about the role of individual weather systems, the larger values over the British Isles and Western Russia
seem consistent with more frequent persistent atmospheric blocks over these regions.

### 4.1.2   Asymptotic methods to characterise variability across timescales

Persistence in a system is associated with memory effects that lead to variability being concentrated at long, rather than short timescales. How variability in the series is distributed across timescales is therefore an important indicator of global persistence and can point to relevant persistence timescales. Specifically, persistence often translates into scaling laws: in the time series
variability as a function of frequency or timescale (for stationarity) (Bunde et al., 2002), but also in the distribution of inter-event





times (for recurrence, see section 5.4). The slope of these laws indicates the degree of persistence. Time series also often exhibit different scaling laws over different frequency intervals, highlighting how persistence may differ across timescales in the data.

A first scaling law can be obtained by considering how autocorrelation decreases as a function of time lag. The more stationary a series, the less rapidly its autocorrelation should decrease. Conceptually, time series can be broadly divided between short-
range and long-range dependence series. For a short-range dependent series (like autoregressive models), the autocorrelation decreases rapidly with the time lag, eventually reaching 0 after a certain lag or decaying exponentially:

$$\rho(k) \sim \alpha^k \tag{5}$$

as $k \to \infty$, and with $0 < \alpha < 1$. By contrast, in a long-range dependent series, the autocorrelation decays following a power-law:

$$\rho(k) \sim k^{2d-1} \tag{6}$$

where $d$ is called the dependence parameter ($0 < d < \frac{1}{2}$) (Beran, 2017; Franzke, 2013). A lower $d$ is associated with a slower decay of the autocorrelation function, hence with more stationarity systems. White noise has $d = 0$. Witt and Malamud (2013) and Franzke (2013) discuss several ways how $d$ can be estimated.

Like autocorrelation $\alpha$ and $d$ (the dependence parameter) measure global stationarity in continuous time series because they are
based on asymptotic relationships. However, they cannot highlight specific stationarity timescales in a series, as autocorrelation does. Applications to atmospheric time-series show that temperature exhibits long-term dependence (e.g., Yuan et al., 2010; Koscielny-Bunde et al., 1998; Eichner et al., 2003), while precipitation exhibits either short- or long-term dependence (Potter, 1979; Jiang et al., 2017; Yang and Fu, 2019). Franzke (2013) computed $d$ and found long-range dependence in North Atlantic winds (Figure 2-b).

The Hurst coefficient (or exponent) $H$ (Hurst, 1951) is a common measure of memory in a time series which is obtained from a second scaling law. Specifically, $H$ characterises how a time series $(X_t)_t$ fluctuates relative to its mean. Noting

$$Z_i^n = \sum_{t=(i-1)n+1}^{in} (X_t - \mathbb{E}[X_t]) \tag{7}$$

the cumulative fluctuations of $(X_t)_t$ relative to its mean, calculated over intervals of size $n$, Hurst argued, empirically from geophysical time series, that these fluctuations could exhibit scale-invariant properties. In other words, cumulative fluctuations
over two different timescales $n$ and $m$ are related through

$$Z_i^n \overset{d}{=} \left(\frac{n}{m}\right)^H Z_j^m \ \forall \ (i,j,k,l) > 0 \tag{8}$$

where $\overset{d}{=}$ stands for equality in distribution. $H$ ranges from 0 to 1. $H < 1/2$ indicates anti-stationary behaviour, such that successive increments of $X_t$ relative to its mean ($X_t - \mathbb{E}[X_t]$) tend to be negatively correlated, and the time series fluctuates substantially at short timescales. By contrast, $H > 1/2$ points to stationary behaviour, in which successive increments are
positively correlated, and variability is concentrated at long timescales. $H = 1/2$ corresponds to white noise (no temporal





correlation). Thus, the higher $H$ is, the smoother the time series. $H$ is also theoretically related to the dependency parameter and the power spectrum exponent (see below) (Franzke et al., 2020).

$H$ can be estimated in many ways (see e.g., Koutsoyiannis, 2003; De las Nieves López García and Requena, 2019; Franzke et al., 2020). In the literature, $H$ has mainly been used to characterise long-term dependence/stationarity (Mandelbrot and Wallis,
1969), for instance in series of monthly- or annual-mean temperature (MacDonald, 1992; Kumar et al., 2013), precipitation (Bunde et al., 2013) or drought indices (Tatli, 2015). However, some studies also calculated it for daily time series (e.g., Rehman and Siddiqi, 2009; Velásquez Valle et al., 2013).

Like the Hurst exponent, spectral analysis also characterises how a time series' variability is distributed across timescales. The power spectrum is commonly defined as the Fourier transform of the autocorrelation function $\rho$:

$$S(f) = \int\limits_{-\infty}^{\infty} \rho(k)e^{-2i\pi f k}dk \tag{9}$$

where $f$ is the frequency and $k$ the time lag. In a pure white noise series, the variability is distributed equally across frequencies. Hence, the power spectral density is a constant. If temporal dependence is present, however, the power spectrum typically exhibits a power-law decrease with frequency $f$:

$$S(f) \sim f^{-\beta} \tag{10}$$

where $\beta$, called the power spectrum exponent, indicates the degree of stationarity in the time series ($0 < \beta < 1$). For statistically stationary series, one can also show that $\beta = 2H - 1$ where $H$ is the Hurst coefficient (Parzen, 1986).

The larger $\beta$ is, the more variance is concentrated at low frequencies. This implies a memory effect at low frequencies that relates to stationarity in the time series. Such scaling behaviour is very common in climatological series, and the spectrum is often divided into distinct scaling regimes corresponding to specific frequency intervals. Fraedrich and Larnder (1993) discuss
the example of precipitation in continental Europe, and relate the different regimes to specific physical processes and timescales (from individual storms at high frequencies to climate fluctuations at low frequencies). Yang and Fu (2019) obtain similar results using hourly and daily precipitation data for the United States. Pelletier and Turcotte (1997) also used power spectra to quantify stationarity in various monthly climatic series, as did Ault et al. (2014) for drought stationarity and MacDonald (1992) for temperature stationarity. Telesca et al. (2016) analysed stationary regimes in 10-minute wind series across Switzerland.

## 395  4.2  State stationarity

We now turn to methods that focus on stationarity of specific system states. Unlike global methods, which take one-dimensional time series as input, several of the following methods are directly applicable to multidimensional data. We begin with methods that identify stationary states from the system trajectory (sections 4.2.1-4.2.4), before discussing methods that quantify the average stationarity of a system state (sections 4.2.5-4.2.6).





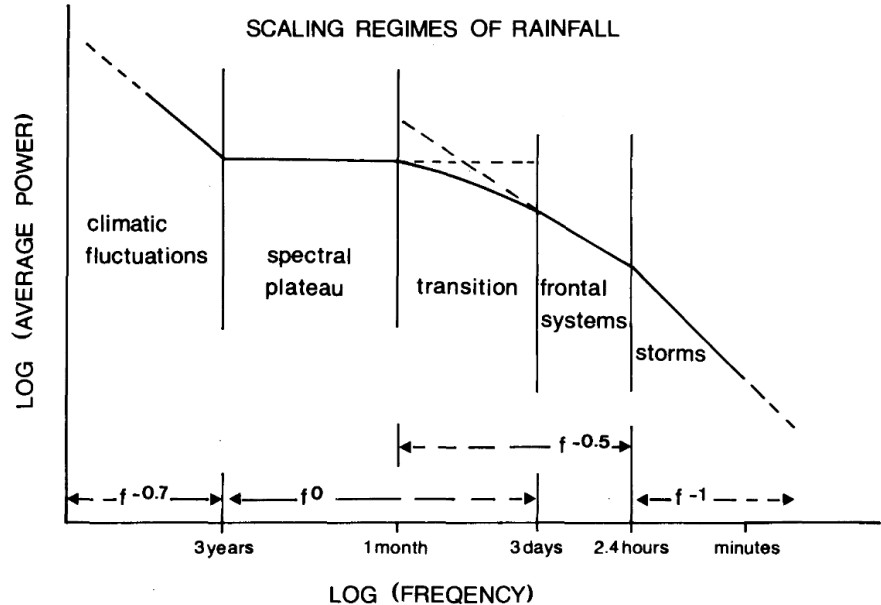

**Figure 3.** Schematic diagram of the scaling regimes of continental European rainfall obtained by spectral analysis, along with the hypothesised meteorological interpretations of the various regimes. Reproduced under the terms of the Creative Commons CC BY license from Fraedrich and Larnder (1993).

### 4.2.1 Weather regimes

We begin with an identification method for stationary states based on the concept of "weather regimes" (Michelangeli et al., 1995). This concept emerges from the realisation that the extra-tropical atmospheric circulation evolves mainly as a succession of a handful of large-scale circulation patterns (Hannachi et al., 2017). These preferred sub-seasonal flow patterns, or weather regimes, tend to be quasi-stationary over timescales of a few days to a few weeks and are therefore strongly related to weather persistence. They account for much of the low-frequency atmospheric variability at intra-seasonal timescales (Pandolfo, 1993; Hannachi et al., 2017). The existence of weather regimes in the mid-latitudes has long been recognised, one of the earliest being atmospheric blocking (e.g., Namias, 1964). They have attracted considerable attention, in particular because of the potential long-range predictability they offer (e.g., Ghil and Robertson, 2002; Büeler et al., 2021) and their link to surface impacts (e.g., Grams et al., 2017).

There are many ways to calculate weather regimes (see Huth et al. (2008) and Hannachi et al. (2017) for detailed overviews). The most common methods include pattern recognition and dimensionality reduction techniques, applied to a proxy variable for the atmospheric circulation like 500 hPa geopotential height or sea-level pressure. Principal Component Analysis is a frequent choice (Fereday, 2017; Grams et al., 2017). Pattern recognition techniques include self-organizing maps (Huth et al., 2008; Francis et al., 2020; Weiland et al., 2021) and clustering algorithms (Figure 4). The latter can be probabilistic (e.g., Gaussian mixture models; Woollings et al., 2010), hierarchical (e.g., Ward clustering; Hannachi et al., 2012) and non-hierarchical (e.g.,





k-means or Partitioning Around Medoids; Grams et al., 2017). Various statistics (gap statistic, silhouette coefficient, etc.) can help objectively select an optimal number of clusters, many of which are available from the R package `clusterCrit` (Desgraupes, 2018). Note that in practice, the input data should be normalised to remove long-term and seasonal trends to focus exclusively on intra-seasonal variability (Grams et al., 2017).

Regimes can also be identified as local maxima in the (multidimensional) PDF of the target field, obtained empirically through e.g., kernel smoothing (Kimoto and Ghil, 1993; Woollings et al., 2010), or with more complex tools of network theory (Mukhin et al., 2022) and topology (Strommen et al., 2022). Finally, Franzke et al. (2011) identify regimes in the North Atlantic jet position with a Hidden Markov Model (HMM). HMMs are a powerful tool that brings together Markov models and Gaussian mixture models. Given $N$ unknown (hidden) states, the HMM models the distribution of the observed series conditionally

on each hidden state, with the sequence of hidden states following a first-order Markov process (Franzke et al., 2008). The transition matrix, hidden states and conditional distributions can be estimated simultaneously.

The main drawback of the weather regime approach is that it is biased toward preferred, or frequent, states. Stationary but rare flow patterns that fall outside the range of the major regimes may therefore be missed. Additionally, certain patterns may fall in between regimes and are not or mis-classified. Still, weather regimes are very useful because they transform complex

multidimensional systems into categorised, one-dimensional series (according to which regime the system is closest to at each time step).

### 4.2.2 Quasi-stationary states

One major disadvantage of weather regimes is that they strongly simplify the phase space. They also focus on the patterns that account for most of the variability in the data, regardless of their persistence. They may thus miss rare yet impact-relevant

persistent patterns. Several other techniques exist to directly extract stationary patterns from the system trajectory.

Stationary circulation patterns can be seen as mathematically quasi-stationary solutions of the atmosphere's equations of evolution (Mo and Ghil, 1987). Such solutions are characterized by average system time derivatives close to zero, meaning that the system tends to remain in their vicinity longer than elsewhere in the phase space – hence their link to stationary states. Stationary states can therefore be directly identified from the system's dynamics by looking for states whose time derivative

is close to zero. If we know the system's exact evolution equations (in the case of simplified models, for example), strictly stationary solutions can be directly computed (e.g., Charney and DeVore, 1979; Legras and Ghil, 1985; Mo and Ghil, 1987). In practice, however, this is rarely the case, and quasi-stationary (or "metastable") states are defined in a statistical sense only, as those for which system tendencies (i.e., time derivatives) approach zero (Vautard, 1990):

$$\{\mathbf{x}^* \text{ such that } \mathcal{T}(\mathbf{x}^*) \approx 0\} \tag{11}$$

$\mathcal{T}(\mathbf{x}^*)$ is the composite tendency at $\mathbf{x}^*$, defined as the average (or area-average for multidimensional fields) of instantaneous tendencies at all occurrences of $\mathbf{x}^*$ – in practice at all times when the system is in the neighbourhood of $\mathbf{x}^*$:

$$\mathcal{T}(\mathbf{x}^*) = \sum_t \left.\frac{d\mathbf{x}}{dt}\right|_{\mathbf{x}(t)} \mathbb{1}\{d(\mathbf{x}^*,\mathbf{x}(t)) \leq d_0\} \tag{12}$$



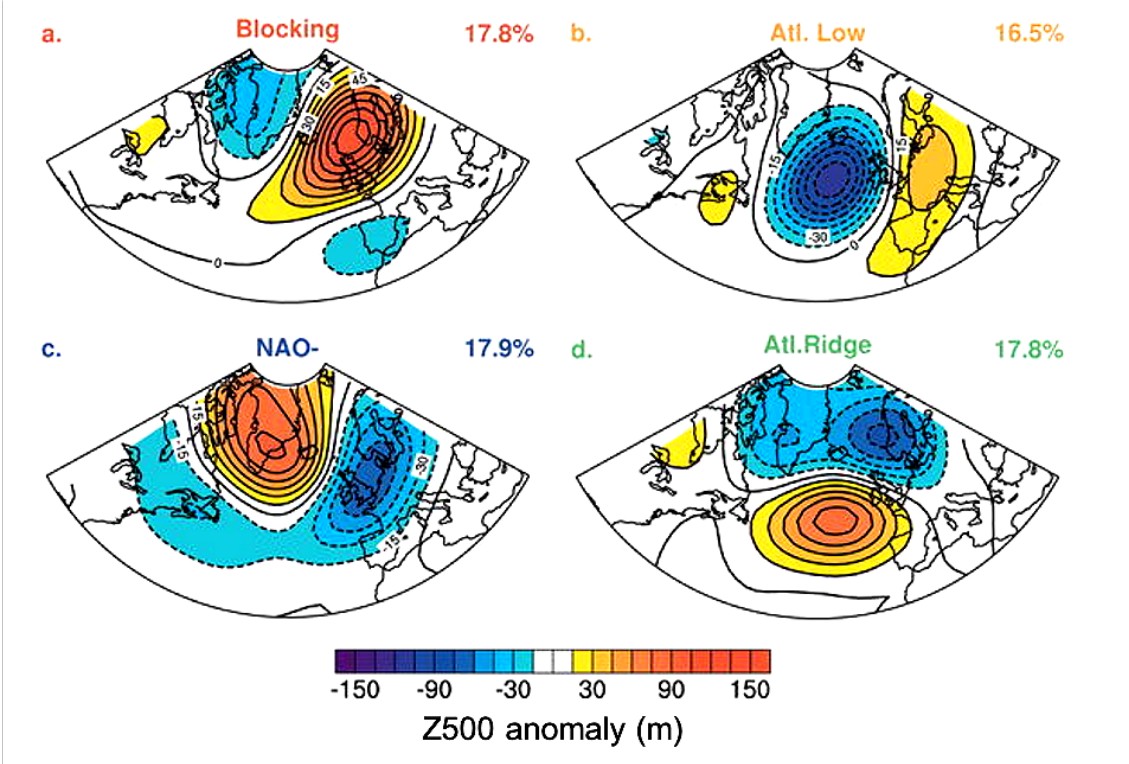

**Figure 4.** Summer weather regimes computed by k-means clustering from daily Z500 fields over the North Atlantic–European sector from 1950 to 2003 (data from NCEP-NCAR reanalysis). To eliminate transient and ambiguous episodes, only sequences of 5 days or more occupied by the same regime are retained (hence total regime frequency summing to $\approx 70\%$). Reproduced with permission from Cassou et al. (2005).

where $d(\cdot, \cdot)$ is a similarity metric and $d_0$ some small threshold. It is also possible to weigh the terms in 12 according to $d(\mathbf{x}^*, \mathbf{x}(t))$ (Vautard, 1990; Michelangeli et al., 1995). When dealing with complex multidimensional systems, it is simpler to

calculate tendencies on the time series of the system's leading principal components. Equation 12 can then more easily be solved by minimising $|\mathcal{T}(\mathbf{x})^2|$ in $\mathbf{x}$, with distinct solutions corresponding to different quasi-stationary states. In practice, instantaneous tendencies at any $\mathbf{x}^*$ can exhibit a large variance, and time series are commonly low-pass filtered to remove the short-term noise that would complicate solving for 12. Additionally, $|\mathcal{T}(\mathbf{x})^2|$ can be difficult to minimise as it is piecewise constant (due to the finite number of observed $\mathbf{x}$ values). Only approximate solutions are achievable. It may thus be difficult to know precisely how

many zeros of $|\mathcal{T}(\mathbf{x})^2|$ exist and whether two approximate solutions correspond to the same minimum. Vautard (1990) presents a method to select relevant solutions.

It is important to note that even if $T_i(\mathbf{x}^*) \approx 0$, instantaneous tendencies $\frac{d\mathbf{x}}{dt}\Big|_{\mathbf{x}(t)}$ in equation 12 may be far from zero (Vautard, 1990). Indeed, the time derivative of $\mathbf{x}(t)$ usually depends not only on $\mathbf{x}$ but also on other time-dependent variables that may evolve independently from $\mathbf{x}$.

Haines and Hannachi (1995) and Hannachi (1997) estimated quasi-stationary states over the North Pacific in the output from a





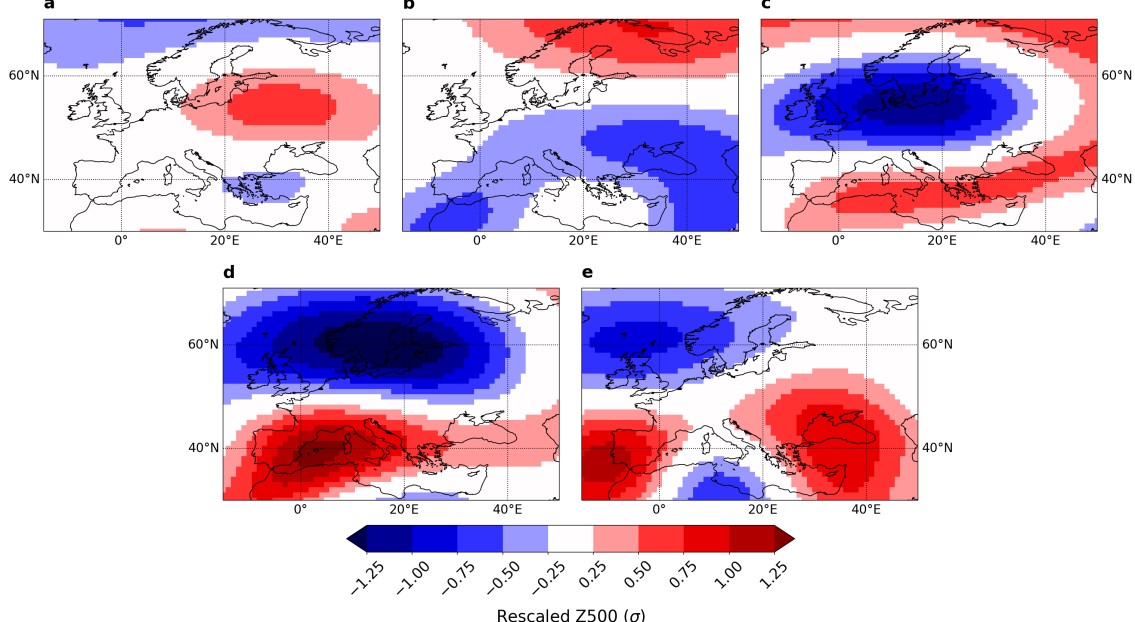

**Figure 5.** Five quasi-stationary states of JJA 5-day averaged rescaled Z500.

global climate forced by perpetual January conditions. Vautard (1990) found four main quasi-stationary patterns in wintertime daily 700 hPa geopotential height fields over the Atlantic, and analysed their stationarity and onset/break characteristics. Michelangeli et al. (1995) also looked at 700 hPa geopotential fields and compared quasi-stationary states with the leading EOFs over the Atlantic and Pacific Oceans during winter. Mo and Ghil (1987) did a similar comparison for the Southern

Hemisphere circulation during the austral winter.

Figure 5 shows five quasi-stationary Z500 anomaly patterns over Europe in summer. We obtained them from the 10 leading EOFs of the 5-day averaged Z500 fields, following Vautard (1990). The patterns were obtained by clustering the resulting approximate solutions with the highest number of clusters for which the resulting patterns were subjectively different. The solutions include blocking-type patterns over Scandinavia/Western Russia (Figure 5-a,b) and zonal patterns (Figure 5-c,d) that

are similar to the results of Figures 4 and 10, and to the canonical patterns of variability over the Euro-Atlantic sector (Grams et al., 2017).

### 4.2.3 Optimally persistent patterns

The technique of optimally persistent patterns (OPPs) was introduced by DelSole (2001). OPP analysis is conceptually similar to principal component analysis; however, instead of patterns that maximise the variance in the observed series, OPPs are defined as

the patterns whose time components (i.e., the projection of the observed series onto the OPP) are the most persistent. Persistence is here measured by the decorrelation time (equation 3) or, alternatively, by the squared decorrelation time $T_2 = 1 + 2 \sum_{k=1}^{N} \rho(k)^2$



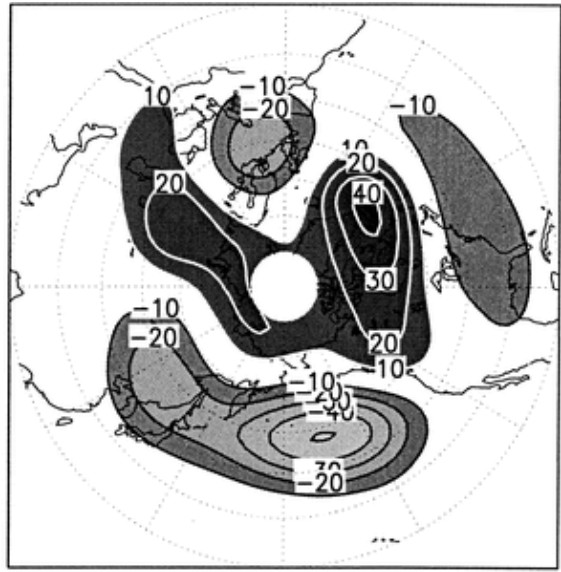

**Figure 6.** The pattern associated with the leading eigenvector maximising the squared decorrelation time for daily anomaly fields of 500 hPa geopotential height for the 1950–1999 period (data from NCEP-NCAR reanalysis; units in m). Adapted from DelSole (2001).

DelSole (2001) (with $\rho(k)$ the autocorrelation function and $N$ the number of observations). If the dynamical system $\mathbf{x}(t)$ is m-dimensional, then for an m-dimensional pattern $\mathbf{u}$, the time component of $\mathbf{u}$ is defined as $y(t) = \mathbf{u}^T\mathbf{x}(t)$. If $\rho_y(k)$ is its associated autocorrelation function, finding OPPs then consists in maximising $1 + 2\sum_{k=1}^{N}\rho_y(k)$ or $1 + 2\sum_{k=1}^{N}\rho_y^2(k)$. In the first

case (maximising the decorrelation time), the optimisation reduces to an eigenvalue problem, details of which can be found in Hannachi (2021). The leading eigenvector maximises the decorrelation time. OPPs can then be obtained by projecting the observed data $\mathbf{x}(t)$ onto the time series associated with each eigenvector. The second case (maximising the squared decorrelation time) leads to a more complex non-linear optimisation problem that can be solved iteratively (see details in DelSole (2001)). Note that as in section 4.2.2, the system should first be embedded into a lower-dimensional space, e.g., by projecting it onto

the set of leading EOFs. Figure 6 shows the leading OPP obtained by DelSole (2001) by maximising the squared decorrelation time for daily Northern Hemisphere Z500 fields.

OPPs are especially relevant for forecasting since they correspond to the patterns with the most low-frequency variability. However, like EOFs and quasi-stationary states, OPPs are mathematical objects that do not necessarily correspond to "real" patterns ever attained by the system.

**4.2.4 Extreme values and dynamical systems theory**

The quasi-stationary state method only focuses on the most stationary states of a system, and OPPs characterise stationarity for a handful of possible patterns only. By contrast, dynamical systems theory provides a convenient framework to describe the stationarity of any system state (Lucarini et al., 2016). In this framework, stationarity is defined for any point $\mathbf{x}_0$ of the



phase space as the inverse of the average residence time of trajectories around $\mathbf{x}_0$. The residence time is calculated based on the
distance between successive system values, with two successive values deemed similar if their distance is below some small
threshold. For any state $\mathbf{x}_0$, the probability that $\mathbf{x}(t)$ will remain in a close neighbourhood around $\mathbf{x}_0$ (a ball of radius $\epsilon$) can be
approximated by (Faranda et al., 2017a, b)

$$\mathbb{P}\left(d(\mathbf{x}(t), \mathbf{x}_0) \leq \epsilon\right) \simeq \exp\left\{-\theta(\mathbf{x}_0) \frac{x - \mu(\mathbf{x}_0)}{\sigma(\mathbf{x}_0)}\right\} \tag{13}$$

where $d(\cdot, \cdot)$ is a distance function (Euclidean distance in Faranda et al. (2017a, b)) and $\theta$ is called the extremal index. $\epsilon$ is
usually chosen to be the $2^{\text{nd}}$ percentile of the $d(\mathbf{x}(t), \mathbf{x}_0)$ values. $\theta$ can be estimated in various ways (Hamidieh et al., 2009;
Holešovský and Fusek, 2022), two common ones being the intervals estimator of Ferro and Segers (2003) and the gap estimator
of Süveges (Faranda et al., 2017a). In dynamical systems theory, stationarity is then defined as $1/\theta$ (some studies directly use
$\theta$ as a measure of stationarity (e.g., Franzke, 2013)). A stationary point of the system (where $\frac{d\mathbf{x}}{dt} = 0$) has infinite stationarity.
By contrast, if trajectories immediately leave the neighbourhood of $\mathbf{x}_0$ then the stationarity is equal to 1. In practice, long
trajectories of $\mathbf{x}(t)$ are required to explore all possible states of the phase space (also called the set of attractor points, or simply
the attractor), and to best approximates sequences of states on the attractor. Note that the choice of $\epsilon$ constrains the values that
$\theta$ can reach. In particular, very small values (necessary for equation 13 to hold) usually lead to $1/\theta$ being around 1-2 days
(e.g., Faranda et al., 2017b, a; Holmberg et al., 2022), so that the resulting persistence metric is highly local in time and not
necessarily relevant for S2S timescales.

The advantage of the method is that it is grounded in mathematical theory and, unlike approaches based on e.g., transition
probabilities, it does not require categorising the data. It also provides an easily interpretable index of stationarity for each
system state (the average residence time of the system's trajectory around that state). Holmberg et al. (2022) have for instance
taken this approach to analyse the link between atmospheric circulation persistence and warm spells in Europe. Faranda et al.
(2019) averaged $\theta$ values across all time steps to quantify global persistence in the North Atlantic flow. Going back to our
example of the European summer circulation, the three major persistent Z500 anomaly patterns are obtained by clustering the
top 3% (115) most persistent daily patterns (Figure 7). We chose 5 cluster centers empirically, after testing for 2-15 centers and
selecting the highest number of clusters for which the resulting patterns were subjectively different. As in Figure 5, persistent
states include blocking regimes (over Scandinavia, Western Russia and the North Sea; Figure 7-a,b,c), a zonal regime (Figure
7-d) and a European ridge regime (Figure 7-e).

### 4.2.5 Residence times

A common way to quantify stationarity in time series relies on the concept of "residence time". Residence times extend the
dynamical systems approach of the previous section. For a continuous series, the residence time $R$ at time $t$ is defined as the
time during which the system remains similar to its value in $t$:

$$R(t) = \max_{k \geq 0}\left\{d\left(\mathbf{x}(t), \mathbf{x}(t+k')\right) \leq \epsilon \ \forall \ 0 \leq k' \leq k\right\} \tag{14}$$



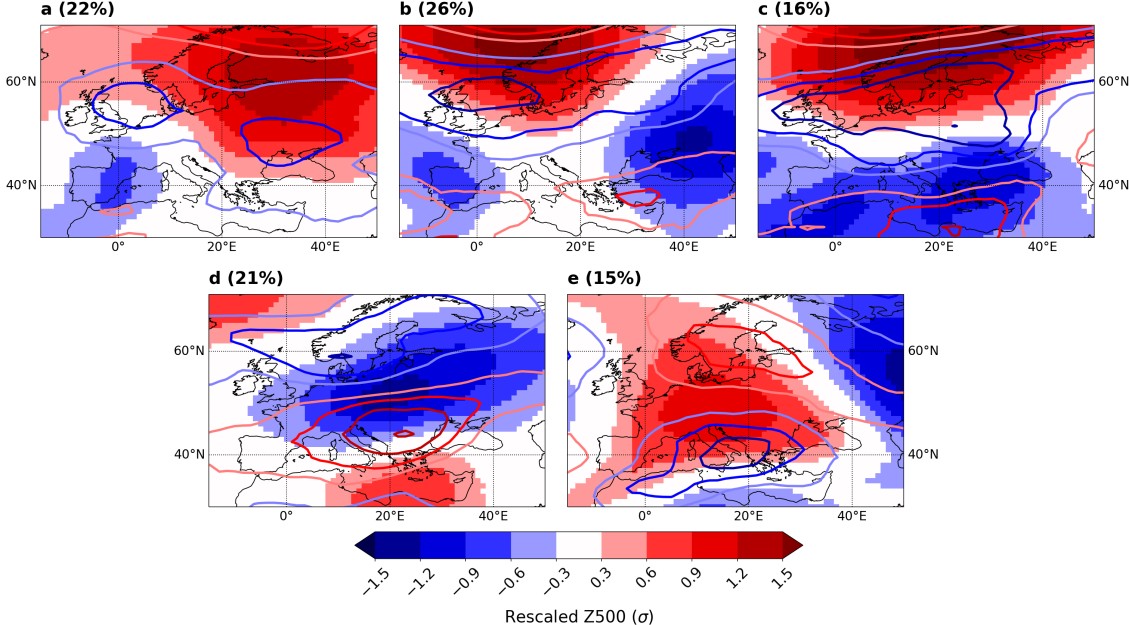

**Figure 7.** Most persistent Z500 anomaly patterns during JJA, obtained by clustering the top 3% (115) daily patterns with the highest stationarity index $1/\theta$. The clustering algorithm is Partitioning Around Medoids (PAM). Also shown are the associated 200 hPa zonal wind anomalies (contours; same levels as Z500). The frequency of each pattern among the 115 most persistent ones is indicated in the top left-hand corner of each panel.

where $d(\cdot,\cdot)$ is a similarity metric and $\epsilon$ is a small threshold. For a categorical series, $R$ is similarly defined as the time during which the system remains in its state $\mathbf{x}(t)$ before transitioning to another state:

$$R(t) = \max_{k \geq 0} \left\{ \mathbf{x}(t+k') = \mathbf{x}(t) \ \forall \ 0 \leq k' \leq k \right\} \tag{15}$$

The definition can be relaxed to allow for brief interruptions in a sequence of similar states (one "average" day between two sequences of warm or wet days, for instance) (e.g., Ali et al., 2021). This definition can be extended to sets $\tilde{\mathcal{S}} \subset \mathcal{S}$ of several system states: $R(t) = \max_{k \geq 0} \left\{ \mathbf{x}(t+k') \in \tilde{\mathcal{S}} \ \forall \ 0 \leq k' \leq k \right\}$ (Richardson et al., 2019). Note that equations 14-15 take a Eulerian perspective; their parallel in the Lagrangian perspective is the concept of "survival time", i.e., the duration of a specific weather system or pattern along its trajectory (Liu et al., 2018; von Lindheim et al., 2021).

The residence time approach can characterise episodic, state, and also global stationarity, and is particularly useful for predictability and risk assessment (De Luca et al., 2019; Francis et al., 2020; Berkovic and Raveh-Rubin, 2022). From the time perspective, large values of $R(t)$ indicate the most stationary periods. A minimum threshold is often defined to separate stationary from non-stationary periods: for instance, 2 days for weather regimes (De Luca et al., 2019; Francis et al., 2020) and extreme precipitation (Du et al., 2022), 5-6 days for warm spells (Berkovic and Raveh-Rubin, 2022; Rousi et al., 2022b), 5-25 days for geopotential anomalies (Dole and Gordon, 1983), or 2 seasons for droughts (Ford and Labosier, 2014).



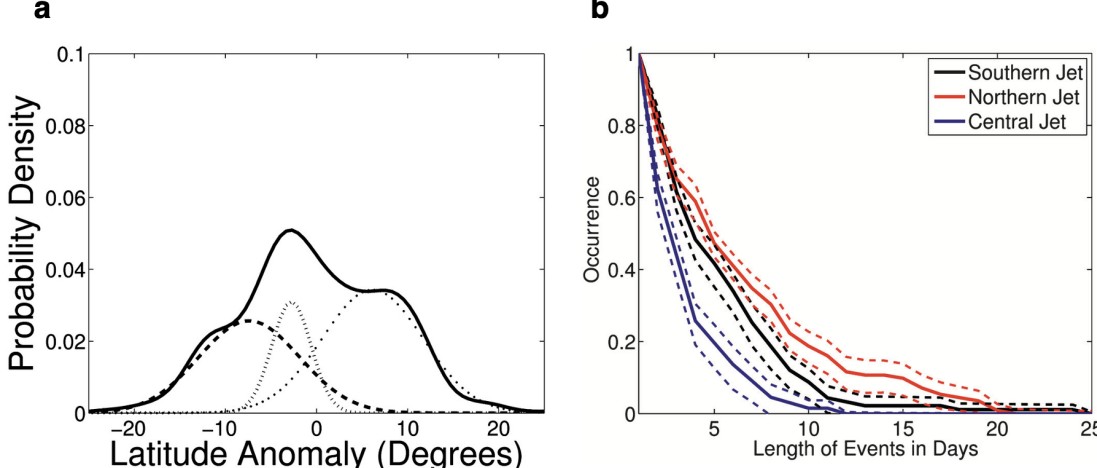

**Figure 8.** PDF of the North Atlantic jet latitude index (solid) together with the weighted Gaussian PDFs from the HMM: the southern (dashed), northern (dotted), and central (dashed–dotted) regimes. Regime duration curves (southern (black), northern (red), and central (blue)), expressed as the frequency of occurrences lasting at least $n$ days. Dashed curves show the corresponding 2.5% and 97.5% confidence levels. Reproduced with permission from Franzke et al. (2011).

In the phase space, the stationarity of any state $\mathbf{x}_0$ can be described from the distribution of its residence times $R_{\mathbf{x}_0} \sim$
$\mathbb{P}\{R(t) \text{ s.t. } \mathbf{x}(t) = \mathbf{x}_0 \text{ and } \mathbf{x}(t-1) \neq \mathbf{x}_0\}$ (Figure 8). The easiest is to calculate the mean (Kyselý and Domonkos, 2006; Kučerová et al., 2017; Richardson et al., 2019), maximum (Rousi et al., 2022b), or some extreme percentile (Zolina et al., 2013) of $R_{\mathbf{x}_0}$. Like autocorrelation, it is also possible to characterise stationarity by the dependence of $\mathbb{P}\left(R_{\mathbf{x}_0} = n\right)$ on $n$ (Sharma and Panu, 2014). Light distribution tails indicate an exponential decay and short-term memory (see section 4.2.6), while heavy tails (e.g., power-law (Bunde et al., 2013) or q-exponential (Weber et al., 2019) distributions) point to long-range dependence in the
system. For example, Pfleiderer and Coumou (2018) fitted exponential models to the distribution of consecutive warm and cold spells to quantify temperature persistence in the Northern Hemisphere, and Huguenin et al. (2020) did the same for weather types over Central Europe. Many other types of distribution can also be fitted (e.g., Deni et al., 2010; Zolina et al., 2013). Residence times can also be modeled as functions of covariates. For instance, Röthlisberger et al. (2019) and Ali et al. (2021) apply a Weibull regression model to relate the duration of dry and wet spells to Rossby wave activity in the mid-latitudes.
Finally, averaging $R(t)$ over time $t$, or $R_{\mathbf{x}_0}$ across all possible system states, provides a convenient and easily interpretable global stationarity metric (e.g., Pfleiderer et al., 2019).



### 4.2.6 Transition probabilities

Residence times characterise how long the system remains in the same state $\mathbf{x}_0$. By contrast, transition probabilities $P$ describe the likelihood that the state of the system does not change at the next time step:

$$P(\mathbf{x}_0) = \mathbb{P}\left(\mathbf{x}(t+1) = \mathbf{x}_0 \mid \mathbf{x}(t) = \mathbf{x}_0\right) \tag{16}$$

Transition probabilities can be further divided by marginal probabilities to yield probability ratios (e.g., Kolstad et al., 2015):

$$PR(\mathbf{x}_0) = \frac{P(\mathbf{x}_0)}{\mathbb{P}\left(\mathbf{x}(t) = \mathbf{x}_0\right)} \tag{17}$$

A high transition probability $P(\mathbf{x}_0)$ or $PR(\mathbf{x}_0)$ means that the system likely remains in $\mathbf{x}_0$, which indicates persistence (Figure 9-a). Like residence times, transition probabilities can be computed on specific system states only, or averaged across all system states (according to their frequency) to yield a measure of global stationarity. Their statistical significance can be estimated by comparing observed transition probabilities to those obtained from a large set of randomly shuffled series. However, unlike residence times, transition probabilities cannot identify stationary periods: they only work in the phase space.

Økland and Lejenäs (1987) applied this method to blocking stationarity; Ford and Labosier (2014), Wilby et al. (2016) and Moon et al. (2018) to droughts; Fereday (2017) to large-scale sea-level pressure patterns; Guilbert et al. (2015) to daily precipitation; and Röthlisberger and Martius (2019) to dry and warm periods (Figure 9-a).

It is often convenient to assume that the distribution of system states at $t+1$ only depends on the system state at $t$, in other words to approximate the system with a first-order Markov chain model (Sericola, 2013). Under this assumption, transition probabilities are directly related to the distribution of residence times. Indeed, if the probability of remaining in state $\mathbf{x}_0$ does not depend on how long the system has previously been in state $\mathbf{x}_0$: $\mathbb{P}\left(\mathbf{x}(t+1) = \mathbf{x}_0 | \mathbf{x}(t) = \mathbf{x}_0\right) = \alpha$, then the probability that $R_{\mathbf{x}_0}$ exceeds $n$ is equal to the probability of finding a sequence of at least $n$ consecutive $\mathbf{x}_0$ values, specifically:

$$\mathbb{P}\left(R_{\mathbf{x}_0} \geq n\right) = \mathbb{P}_t(\mathbf{x}(t-1) \neq \mathbf{x}_0; \mathbf{x}(t) = \mathbf{x}_0; \mathbf{x}(t+1) = \mathbf{x}_0; ...; \mathbf{x}(t+n-1) = \mathbf{x}_0) \tag{18}$$

where the subscript in $\mathbb{P}_t$ specifies that the probability is taken with respect to the time variable, whereas in $\mathbb{P}\left(T_{\mathbf{x}_0} \geq n\right)$ it is taken with respect to the ensemble of residence times for $\mathbf{x}_0$. Thanks to the first-order Markov assumption, equation 18 factors as

$$\mathbb{P}\left(R_{\mathbf{x}_0} \geq n\right) = \mathbb{P}_t\left(\mathbf{x}(t-1) \neq \mathbf{x}_0; \mathbf{x}(t) = \mathbf{x}_0\right) \times$$
$$\prod_{k=0}^{n-2} \mathbb{P}_t\left(\mathbf{x}(t+k+1) = \mathbf{x}_0 | \mathbf{x}(t+k) = \mathbf{x}_0\right) \tag{19}$$

Thus,

$$\mathbb{P}\left(R_{\mathbf{x}_0} \geq n\right) = \beta \times \alpha^{n-1} \tag{20}$$

where $\beta = \mathbb{P}_t\left(\mathbf{x}(t-1) \neq \mathbf{x}_0; \mathbf{x}(t) = \mathbf{x}_0\right)$ does not depend on $t$ for a stationary system, and

$$\mathbb{P}\left(R_{\mathbf{x}_0} = n\right) = \beta(\alpha-1)\alpha^{n-2} \tag{21}$$



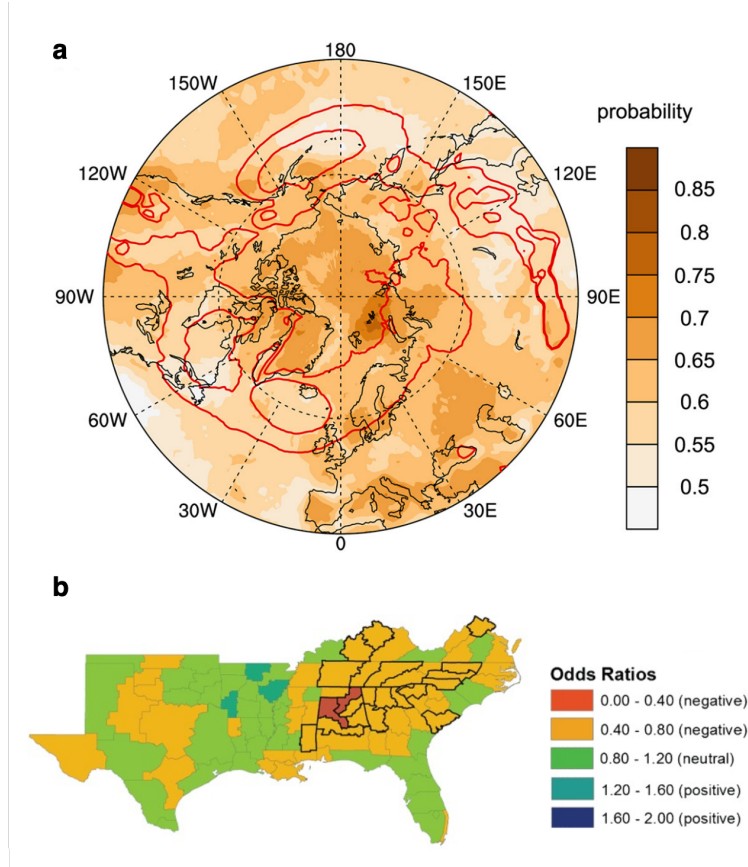

**Figure 9.** (a) Climatological May–October (MJJASO) hot spell survival probabilities (shaded) in ERA-Interim reanalysis data. Red contours show MJJASO cyclone frequencies of 30% and 40%, respectively. Reproduced with permission from Röthlisberger and Martius (2019). (b) Change in the odds ratio of drought occurrence in spring given a 1-unit increase in winter standardised precipitation index, estimated from logistic regression. Thick black contours indicate statistical significance with 95% confidence. Adapted with permission from Ford and Labosier (2014).

The logarithm of the distribution of residence times is therefore a linear function of $n$, with the slope ($\log(\alpha)$) equal to the transition probability. The Markov assumption is thus consistent with a short-term dependent system, in which residence time probabilities decay exponentially with $n$ (see section 4.2.5 and Bunde et al. (2013)). For instance, Huguenin et al. (2020) fit such exponential laws and use their slope as measures of weather regime stationarity over Central Europe in current and future climates. This approach is well-suited to comparing two different series (e.g., two different periods or locations) because it
separates changes in the marginal frequency of state $\mathbf{x}_0$ (only affecting $\beta$) from changes in stationarity (only affecting $\alpha$).





### 4.2.7 Time series modeling

Another possibility to investigate state stationarity is to fit a statistical model to the data that links successive system values with each other:

$$\mathbf{x}(t+1) \stackrel{d}{=} \mathcal{F}\left(\{\mathbf{x}(k)\}_{k=1..t}, \{\theta(t)\}_{k=1..t}\right) \tag{22}$$

where $\stackrel{d}{=}$ stands for equality in distribution and $\{\theta(t)\}_{k=1..t}$ are covariates. Transition probabilities (section 4.2.6), for example, are equivalent to modeling a system's evolution by a transition matrix between all possible system states. Hidden Markov Models (section 4.2.1) likewise estimate transition matrices between hidden states. Logistic regression can be used to assess stationarity in a given state. Several studies used it to quantify stationarity in droughts, using previous precipitation or soil moisture anomalies as predictor variables ($\theta(t)$ in equation 22) (Ford and Labosier, 2014; Meng et al., 2017) (Figure 9-b). The

link to drought occurrence at the previous time step is implicitly taken into account with previous precipitation or soil moisture anomalies. If previous system values are explicitly included as covariates to the model, one speaks of autologistic regression (Wolters, 2017).

### 4.3 Running window methods for episodic persistence

The running window approach identifies persistent periods of a given fixed length in continuous or categorical data. It assesses

the degree of stationarity over fixed time intervals by means of a "similarity index". Given a time interval $\mathcal{T} = \{t_1, ..., t_n\}$ of length $n$ and a similarity metric $d(\cdot, \cdot)$, the similarity index $S_n(\mathcal{T})$ is defined as the average similarity between system values in $\mathcal{T}$. This similarity can be computed with reference to one time step $\mathbf{x}(t_k)$ in particular (typically the first one):

$$S_n(\mathcal{T}) = \frac{1}{n-1} \sum_{\substack{1 \le k' \le n \\ k' \ne k}} d\left(\mathbf{x}(t_{k'}), \mathbf{x}(t_k)\right), \tag{23}$$

or it can measure the average similarity across all pairs of values:

$$S_n(\mathcal{T}) = \frac{2}{n(n-1)} \sum_{\substack{1 \le k, k' \le n \\ k \ne k'}} d\left(\mathbf{x}(t_k), \mathbf{x}(t_k')\right) \tag{24}$$

Different time intervals can then be ranked according to their degree of stationarity. Mo and Ghil (1987), for instance, use pattern correlation as similarity metric and set a threshold of 0.5 to separate stationary from non-stationary periods. Another possibility is to keep the $N$ periods with the largest $S_n(\mathcal{T})$ (i.e., to use a percentile-based threshold). Stationary states can then be detected with e.g., pattern recognition techniques applied to the most stationary periods (Mo and Ghil, 1987). We show

an example for summertime European circulation in Figure 10. The most common are blocking-type patterns over Northern Europe/Western Russia (Figure 10-a,b,c), but we also find two pronounced zonal patterns with a southward/northward shifted jet (Figure 10-d,e).

Averaging $S_n(\mathcal{T})$ values across multiple periods of length $n$ can additionally provide for a measure of global stationarity, as did Hoffmann et al. (2021) to analyse 10-day persistence in summer atmospheric circulation. When working with categorical data,



similarity indices can be averaged for each system state to highlight the most stationary ones (Horel, 1985b).

The running window approach is well-suited to compute temporal trends in stationarity, since the similarity index can be defined at all time steps (Hoffmann et al., 2021). One limitation, however, is that results for the same system but calculated for different period lengths are not directly comparable since the marginal distribution of $S_n(\mathcal{T})$ depends on $n$. Furthermore, $S_n(\mathcal{T})$ only measures the average similarity between successive values. A high $S_n(\mathcal{T})$ does not guarantee high stationarity; there could be

breaks in between sequences of similar values. This is especially relevant for long periods during which strict persistence is unlikely to occur.

    While we try to give a comprehensive view of common methods used in stationarity analysis, we could not include all the methods that exist in the literature. Kornhuber and Tamarin-Brodsky (2021), for instance, define stationarity in a Lagrangian context as the zonal velocity of individual weather systems (in their case, localised temperature anomalies), while Hoskins

and Woollings (2015) use Rossby wave phase speed as a proxy for weather stationarity in the mid-latitudes. Finally, in the framework of dynamical systems theory, stationarity can also be equated to the system remaining for some period of time in a small subset of the phase space. During that period, the system may explore different configurations, but fewer ones than if it had been able to evolve freely across the whole phase space (Fereday, 2017). In this sense, persistent states are states from which the system takes (statistically) a long time to reach the rest of its phase space. This perspective relates to the intransitivity

theory of Lorenz (1990) which postulates that the evolution of the atmospheric circulation on S2S timescales can, under certain conditions, be governed by a few well-separated attractors, each with their own preferred states (Weiland et al., 2021).





| Method | Section | Type | Definition | Features | Main limitations |
|---|---|---|---|---|---|
| Autocorrelation | 4.1.1 | Global | Measure of strength of linear dependence between successive system values; equation 1 | Lag-1 autocorrelation frequently used, but can also highlight relevant timescales of self-dependence in a series | Requires 1D data; characterises the strength of the dependence, not "similarity" per se |
| Dependence parameter | 4.1.2 | Global | Asymptotic characterisation of how autocorrelation evolves with time lag | Discriminates short- and long-range dependence | Requires 1D data; based on autocorrelation which characterises the strength of the dependence, not "similarity" per se |
| Hurst exponent | 4.1.2 | Global | Asymptotic characterisation of how cumulative fluctuations of a series around its mean evolve with timescale | Simple metric to identify stationary/anti-stationary/noise-like behaviour in time series | Requires 1D data |
| Power spectrum exponent | 4.1.2 | Global | Asymptotic characterisation of the distribution of time series variability across timescales | Relies on the power spectrum, for which many methods are available | Requires 1D data; power spectrum linked to autocorrelation which characterises the strength of the dependence, not "similarity" per se |
| Weather regimes | 4.2.1 | State | Common method to identify patterns accounting for high variability in multi-dimensional series | Many techniques available; unsupervised method; often good physical interpretation | Requires choosing a number of regimes; strong simplification of the phase space, i.e. rare yet relevant system states may be missed; indirect link to persistence, which requires further analysis |



| Name | Section | Type | Description | Advantages | Disadvantages |
|---|---|---|---|---|---|
| Quasi-stationary states | 4.2.2 | State | Identifies stationary system states (time derivative ≈ 0) | Unsupervised method; best applied to low-dimension systems (dimension reduction required) | Challenging implementation; true number of states can be difficult to assess; stationarity constraint is strong (zero tendency) and highly local in time; no guarantee that resulting states are "physical" |
| Optimally persistent patterns | 4.2.3 | State | Identifies most autocorrelated patterns in multi-dimensional series and associated timescales | Unsupervised method; relevant for forecasting | Implementation can be challenging; limited number of resulting states and no guarantee that they are "physical" |
| Dynamical systems approach | 4.2.4 | State | Defines a persistence index for every point of a system's trajectory based on local residence times | Unsupervised method, well-grounded in mathematical theory; persistence index directly interpretable; works with continuous multi-dimensional data | Based on extreme value approximations; the metric is consequently highly local in time, and may not be relevant at S2S timescales |
| Residence times | 4.2.5 | Global/ State/ Episodic | Defined as the period during which a system remains similar to its initial value | Versatile and easy to interpret approach; residence times can be calculated across time or system states, on continuous or categorical data | Potentially difficult to summarise the distribution of residence times for a given system state |
| Transition probabilities | 4.2.6 | State | Probability that a system remains in the same state at the next time step | Easy to calculate and interpret; directly related to residence times under the Markov assumption | Requires categorical data; limited to one-step-ahead stationarity |
| Time series modeling | 4.2.7 | State | Statistical modeling of successive system values | Potentially very versatile approach that can include the effect of external covariates | Models may capture "dependence" rather than "similarity", and can quickly become complex and difficult to interpret |



| Running window methods | 4.3 | Defines a stationarity value | Episodic for all time steps at a given timescale | Results can be used to identify stationary states | Relative stationarity metric only; difficult to compare different systems |
|---|---|---|---|---|---|

Table 1: Overview of stationarity methods: method name; corresponding section; type of persistence it applies to; brief definition; main features; main limitations.





**Figure 10.** (a-e) Most persistent 10-day averaged Z500 anomaly patterns during JJA, obtained by clustering the top 10% (40) 10-day patterns with the highest similarity index (equation 24). Similarity is measured with the SSIM index. The clustering algorithm is Partitioning Around Medoids (PAM). The associated 200 hPa zonal wind anomalies are shown by contour lines (same levels as Z500). The frequency of each pattern is also indicated. (f-j) 10-day averaged surface temperature anomalies associated with each persistent Z500 pattern.



## 5   Recurrence

We now discuss methods to capture recurrence in atmospheric data (see Table 2 for an overview). In contrast to stationarity, most methods here only quantify recurrence in a specific system state. They typically require binary time series as input (where "1" represents an occurrence of the system state or event of interest) and count event occurrences over time. Recurrence is therefore always defined for a given state or event, usually with the potential for large impacts like tropical (Mumby et al., 2011) and extra-tropical cyclones (Mailier et al., 2006; Vitolo et al., 2009), windstorms (Khare et al., 2015), or extreme precipitation episodes (Kopp et al., 2021). For this reason, we did not explicitly divide this section into global, state and episodic methods. Common methods are available to characterise episodic persistence (section 5.1) and the persistence of given states (sections 5.2-5.5). Few definitions of "global recurrence" exist, however. Additionally, no study has, to our knowledge, tried to analyse S2S recurrence in a "non-supervised" way, i.e. that would allow to objectively identify recurrent states. In section 5.6, we discuss the potential of recurrence plots to fill that gap.

Because most recurrence methods work directly with binary time series, point processes are a convenient theoretical tool in recurrence analysis. Point processes are a class of probability models for the random occurrence of points in a space (one- or multi-dimensional). The most simple hypothesis than can be made is that of complete serial randomness, i.e., points occur completely independently of one another. In this case, recurrence occurs by chance only. The homogeneous Poisson point process is a simple model without memory for binary series in which events occur without correlation and with a constant intensity. In a homogeneous Poisson process, the number of points in individual time intervals are independent, and the number of points in an interval of length $\tau$ is Poisson distributed with rate $\lambda\tau$, where $\lambda$ is the process intensity rate. This model is useful to simulate binary series with complete serial randomness, and therefore provides an empirical basis to assess the statistical significance of various recurrence metrics in observations.

Separating distinct occurrences of a state, pattern or event is critical to distinguish recurrence from stationarity. Over short periods, like a season, it is in principle possible to visually separate multiple systems occurring in close succession, for instance with Hovmoeller diagrams (e.g., Tuel et al., 2022a; Rousi et al., 2022a). Most of the time, however, separation cannot be done by hand, and objective criteria are required. One option are physically-based detection/tracking algorithms – e.g., for cyclones (Hodges, 1995; Wernli and Schwierz, 2006) or atmospheric blocks (Schwierz et al., 2004; Steinfeld, 2021) – that can separate distinct weather systems. A second option is to use a minimum duration between events to classify them as distinct, with the choice depending on the underlying physical system or on impact considerations. Barton et al. (2016), Kopp et al. (2021) and Tuel and Martius (2021a), for instance, require a minimum of 2 days with non-extreme precipitation between extreme precipitation episodes (a typical timescale for extratropical cyclones). In Pfleiderer et al. (2019), a single day is used to separate distinct dry or warm periods.

### 5.1   Window counts

A simple way to identify recurrent periods is to look for periods with high event counts. One possibility is to set a time window and require the number of event counts during this window to exceed some threshold (2 or higher; Figure 11-a). Kopp et al.



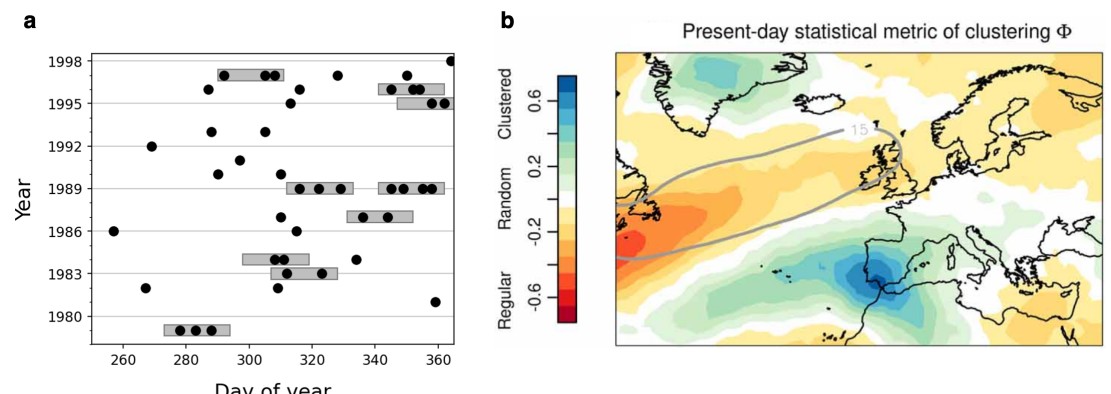

**Figure 11.** (a) Time series of extreme precipitation events (black dots) and 21-day periods containing more than 2 events (gray shading). Adapted from Kopp et al. (2021) under the terms of the Creative Commons Attribution 4.0 License. (b) Map of dispersion statistic $\psi$ (equation 25) of DJF cyclones over the 1975–2004 period, averaged across an ensemble of CMIP5 models. Reproduced from Bevacqua et al. (2020) under the terms of the Creative Commons Attribution 4.0 license.

(2021) and Tuel and Martius (2022a) take this approach to analyse recurrent extreme precipitation episodes at the 2- and 3-week timescales. Pinto et al. (2014) likewise identify cyclone clusters over the British Isles as periods with 4 or more consecutive cyclones within 7 days. Another possibility is to look for clusters of events in which each event is separated by at most $n$ time steps from the preceding one. For instance, Bevacqua et al. (2020) define cyclone clusters as sequences of $N \geq 2$ consecutive cyclones transiting within a given area and separated by 24 hours or less. This method is largely impact-driven and does not aim at modeling the binary series, assessing the significance of the clustering or calculating return periods for large event counts (Dacre and Pinto, 2020). For this, further steps must be taken.

## 5.2 Dispersion metrics

Dispersion metrics characterise the distribution of event occurrences in a time series. In a series with no tendency to clustering, high event counts over a given duration should be much less frequent than in a series where clustering is prevalent. If events occur with the same average probability in the two series, then in the clustered series, sequences with no events should mechanically be more frequent, and the variance in event counts will be higher. In a homogeneous Poisson process (see above), $N_k(\tau)$, event counts in disjoint intervals of length $\tau$ (indexed by $k$), follow a Poisson distribution with parameter $\lambda\tau$. Hence, the expected value $\mu(\tau)$ and variance $\sigma(\tau)^2$ of $N_k(\tau)$ are both equal to $\lambda\tau$. Two common statistical dispersion metrics quantify deviations from this homogeneous behaviour. The dispersion statistic is a measure of temporal correlation over different timescales. It is defined as (Mailier et al., 2006)

$$\psi(\tau) = \frac{\sigma(\tau)^2}{\mu(\tau)} - 1 \tag{25}$$



$\psi$ is related to the Fano Factor ($FF(\tau) = \psi(\tau) + 1$) (Telesca, 2007), also called index of dispersion (Mailier et al., 2006). A positive $\psi(\tau)$ ($\sigma(\tau)^2 > \mu(\tau)$) indicates over-dispersion: events tend to occur in clusters. A negative $\psi(\tau)$ points to under-dispersion: events occur more regularly than in a random process. Near-zero values are consistent with a homogeneous Poisson process (Figure 11-b).

Similarly, the Allan Factor (AF) is defined as the variance of event counts over successive intervals of length $\tau$ divided by twice the average event count in $\tau$ steps (Telesca, 2007):

$$AF(\tau) = \frac{\mathbb{E}[(N_{k+1}(\tau) - N_k(\tau))^2]}{2\mu(\tau)} \tag{26}$$

For a homogeneous Poisson process, $AF(\tau) = 1$, while $AF(\tau) > 1$ indicates clustering behaviour. $\psi$ and $AF$ can be evaluated for a single window $\tau^*$, but assessing how they evolve with $\tau$ helps better detect recurrent behaviour in the series (Serinaldi and Kilsby, 2013). In the absence of clustering, $\psi(\tau)$ and $AF(\tau)$ remain constant with $\tau$. In clustered processes, however, both metrics scale with $\tau$, often with a power-law behaviour over some range $[\tau_1, \tau_2]$ which can be described by an exponent (Telesca, 2007), similar to the spectral analysis of persistence (section 4.1.2). One advantage of $\psi$ and $AF$ is that they are insensitive to the underlying event frequency, which makes them convenient to compare different locations or event identification algorithms with each other (Dacre and Pinto, 2020). However, the corresponding null hypothesis – that the absence of recurrence is equivalent to exponentially distributed event return times under the Poisson assumption – is not always relevant, and other approaches may be better suited to detect deviations from memoryless processes (Blender et al., 2015).

The statistical significance of $\psi(\tau)$ and $AF(\tau)$ can be assessed empirically through Monte-Carlo sampling of random series without memory, or analytically. Mailier et al. (2006) discuss a chi-square test for $\psi(\tau)$ (for a fixed window $\tau$), while Serinaldi and Kilsby (2013) introduce the sampling distribution of the empirical $AF$ estimator. Note that seasonality in event counts can artificially inflate the dispersion statistic, so care must be taken either to remove the seasonality (e.g., Tuel and Martius, 2021a), or to simulate random processes with the same seasonality as the observed signal ($\lambda$ function of the month, for instance). Note also that $\psi(\tau)$ and $AF(\tau)$ both involve $\mu(\tau)$ in the denominator; in other words, these metrics are normalised by the marginal event frequency. The same $\psi(\tau)$ can be obtained for series with different event frequencies, which makes dispersion metrics less relevant for impacts than other methods (e.g, window counts; section 5.1).

Since its introduction by Mailier et al. (2006), the dispersion statistic has been frequently applied in recurrence analysis, namely to detect temporal/serial clustering in cyclones, both tropical (Mumby et al., 2011; Wolff et al., 2016) and extratropical (Vitolo et al., 2009; Pinto et al., 2013, 2014; Economou et al., 2015; Pinto et al., 2016). The Allan factor has been applied to daily precipitation series (Serinaldi and Kilsby, 2013), to wave storms (Besio et al., 2017)and to extreme wind speeds in Switzerland (Telesca et al., 2020).

Alternative metrics that characterise the distribution of event counts have been proposed in the literature. For example, Kopp et al. (2021), working with extreme precipitation counts over 2- to 4- week timescales, proposed a new metric, $S_{cl}(\tau)$ – which they named the "clustering metric" – based on a weighted sum of window event counts. Like the dispersion statistic and Allan factor, $S_{cl}(\tau)$ requires choosing a window size $\tau$. Clustering periods of length $\tau$ are then identified and ranked by applying a moving window sum to the binary series. The period with the most events is selected first; the corresponding events are removed




from the series (to avoid any intersection between clustering periods); and the procedure continues until a minimum number of events is obtained, or until no more clustering periods (with at least two events) are found. This yields a set of $K$ periods with decreasing event counts $n_k(\tau)$: $n_1(\tau) \geq n_2(\tau) \geq ... \geq n_K(\tau)$. Kopp et al. (2021) then define their clustering metric by:

$$S_{\mathrm{cl}}(\tau) = \sum_{1 \geq k \geq K} n_k(\tau) q_k \tag{27}$$

where $q_k$ are weights that can be defined in various ways, as long as they verify (i) that $q_k$ decreases with $k$ (periods with fewer events are given less weight) and (ii) that $q(i) - q(i+1) > q(i+1) - q(i+2)$. In the end, this approach is equivalent to a scoring system for clustering periods, and the resulting metric $S_{\mathrm{cl}}(\tau)$ correlates well with the dispersion statistic.

### 5.3 Ripley's K function

Given a system state $\mathbf{x}_0$, Ripley's K function measures the average number of occurrences of $\mathbf{x}_0$ within a given neighbourhood
around a random occurrence of $\mathbf{x}_0$. It was originally developed for spatial data, but it can be applied in a temporal setting to characterise recurrence in $\mathbf{x}_0$. Let the binary variable $Y_t = \mathbb{1}\{X_t \text{ is in state } \mathbf{x}_0\}$; Ripley's K function for a neighbourhood of size $n$ is then equal to (Ripley, 1981; Dixon, 2014):

$$K(n) = \lambda^{-1} \left( \mathbb{E}\left[ \sum_{k=-n}^{n} Y_{t+k} \mid Y_t = 1 \right] - 1 \right) \tag{28}$$

where $\lambda = \mathbb{E}[Y_t]$ (frequency of $\mathbf{x}_0$ in $X_t$). The statistical significance of $K(n)$ can then be assessed by comparing it to values
obtained for a homogeneous Poisson process of same length and with the same average rate of occurrence as $Y_t$ (Figure 12). In equation 28, $\lambda^{-1}$ can also be discarded, so that $K(n)$ represents an average event count and is more directly related to impacts. Ripley's K was used in meteorology to characterise S2S recurrence in extreme precipitation events by Barton et al. (2016), Tuel and Martius (2021a) and Tuel and Martius (2021b).

### 5.4 Distribution of inter-event times

735 Stationarity in a given system state $\mathbf{x}_0$ can be described by the corresponding distribution of residence times (section 4.2.5). Similarly, the distribution of inter-event times (also called "recurrence times" or "waiting times") characterises the recurrence in $\mathbf{x}_0$ from a global perspective (Altmann and Kantz, 2005). The recurrence time $T_i$ is the duration between the $i$-th and $(i+1)$-th occurrences of state $\mathbf{x}_0$ in a time series. We can easily see that both short and long recurrence times will be more frequent in a series that exhibits clustering than in a series where events occur independently of one another. The presence of a heavy tail in
the distribution of $T_i$ therefore points to a tendency for temporal clustering in the series. If $\mathbf{x}_0$ is rare enough, the distribution of $T_i$ may converge to well-known distributions like the exponential (a sign of short-term dependence, as with homogeneous Poisson processes), or the stretched exponential or power-law distribution (Eichner et al., 2007; Corral, 2015). When events correspond to extremes above a given threshold, the exponents that characterise these distribution are even related to the autocorrelation coefficient of the original series (Santhanam and Kantz, 2008; Kalra and Santhanam, 2021).





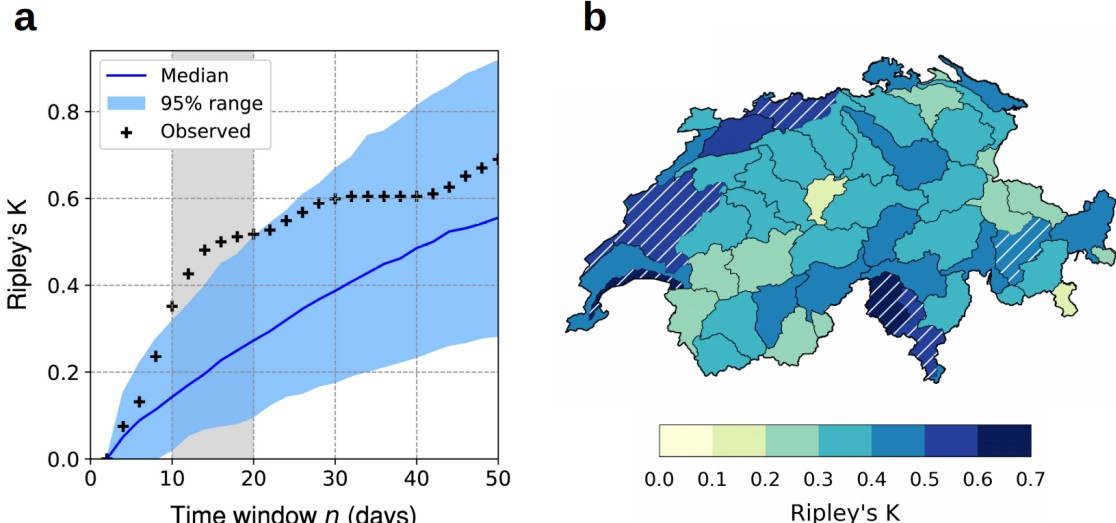

**Figure 12.** (a) Assessing the significance of clustering with Ripley's K function: conceptual example of Ripley's K values on an observed series (black crosses) and the corresponding 95% uncertainty range obtained from random simulations of series with no clustering (blue shading). In this case, observations show significant clustering at the 10-20 day timescale. Reproduced from Tuel and Martius (2021a). (b) Map of 20-day Ripley's K values for catchment-averaged extreme precipitation during fall in Switzerland. Hatching indicates statistical significance at the 95% level. Adapted from Tuel and Martius (2021b) under the terms of the Creative Commons Attribution 4.0 license.

### 5.5 Stochastic modeling of event series

We have so far discussed diagnostic methods that characterise recurrence in binary time series. A different approach consists in fitting point processes to the series. A point process is a random process representing the occurrence times of specific events. It can be characterised by event times $T(n)$ (time of the n-th event in the series) or by the distribution of event counts $N(t_1, t_2)$ for any interval $[t_1, t_2]$. We can define its *intensity* or *hazard* function $\lambda(t)$ by

$$\lambda(t) = \lim_{\Delta_t \to 0^+} \frac{1}{\Delta t} \mathbb{P}(N(t, t + \Delta t) = 1) \tag{29}$$

and, for any $t_1 < t_2$, its *intensity measure* $\Lambda(t_1, t_2)$ as

$$\Lambda(t_1, t_2) = \int_{t_1}^{t_2} \lambda(t) dt \tag{30}$$

Many different point processes have been described in the statistics literature (e.g., Cox and Isham, 1980), but clustering analyses in weather/climate studies have mainly relied on Poisson processes. In a Poisson process, $\lambda(t)$ is a deterministic function of time. If $\lambda(t)$ is constant, the process is said to be homogeneous; otherwise, the process said to be inhomogeneous. For any $t_1 < t_2$, $N(t_1, t_2)$ follows a Poisson distribution with rate $\Lambda(t_1, t_2)$:

$$N(t_1, t_2) \stackrel{d}{=} \text{Poisson}(\Lambda(t_1, t_2)) \tag{31}$$



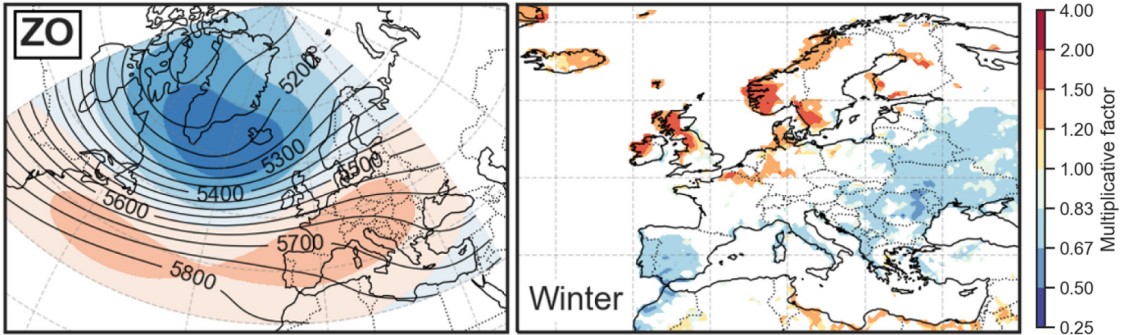

**Figure 13.** Effect of Euro-Atlantic zonal regime occurrence (corresponding 500 hPa geopotential mean and anomalies shown on left panel) on the temporal clustering of extreme precipitation at the three-week timescale during winter (measured as a probability multiplicative factor). Reproduced from Barton et al. (2022) under the terms of the Creative Commons Attribution 4.0 license.

and event numbers in disjoint intervals are independent from each other.

The point process approach has mainly been used to relate recurrence in extreme events to atmospheric or climate variability. To that end, $\lambda(t)$ is assumed to depend on given time covariates $X(t)$. A linear dependence corresponds to Poisson Generalised Linear Models (GLMs):

$$\lambda(t) = \beta \cdot X(t) \tag{32}$$

For instance, Villarini et al. (2011), Tuel and Martius (2022a) and Tuel and Martius (2022b) used Poisson GLMs to relate temporal clustering in heavy precipitation to large-scale modes of climate and atmospheric variability. Mailier et al. (2006) did the same for extratropical cyclones in the North Atlantic. The linear constraint can be lifted by using General Additive Models, or GAMs, in which the relationship between $\lambda(t)$ and the covariates is specified by flexible smooth functions (Hastie, 1992). Villarini et al. (2013) used GAMs to relate flood recurrence in the US Midwest to modes of climate variability like ENSO, and Barton et al. (2022) used GAMs to link temporal clustering in extreme precipitation across Europe to weather regimes in the North Atlantic. Another extension of the Poisson GLM is the Cox regression model (Smith and Karr, 1986), in which

$$\lambda(t) = \lambda_0(t) \exp(\beta \cdot X(t)) \tag{33}$$

where $\lambda_0(t)$ is a baseline intensity/hazard function and $\exp(\beta \cdot X(t))$ accounts for the time-varying effects of the selected covariates. Examples include Villarini et al. (2013), Mallakpour et al. (2017) and Yang and Villarini (2019), who applied Cox regression to recurrence in floods and extreme precipitation. More complex models are also possible; for instance, Khare et al. (2015) introduced a Poisson mixture model for windstorm clustering, in which $\lambda(t)$ is expressed as a stochastic function of time modulated by a gamma distribution (Khare et al., 2015). Note that models can be fitted directly by maximising the likelihood function (Yang and Villarini, 2019) or indirectly by working with window event counts (Mailier et al., 2006; Tuel and Martius, 2022a; Barton et al., 2022).





### 5.6 Recurrence plots

All previously discussed methods share the same limitation: they require binary time series as input, meaning that they are
designed to characterise recurrence only in a given system state. It is not possible with such methods to explore recurrence
in an "unsupervised" way, i.e., to automatically detect recurrent system states or sequences of states, and associated recurrent
periods. Recurrence plots (RPs) may offer a solution to this problem.

An RP is a graphical representation of a system's self-similarity with time (Marwan et al., 2007). Mathematically, it is defined
as a two-dimensional binary matrix $\mathbf{R} \in \{0,1\}^{N \times N}$, with time on both axes, that consists of pairwise comparisons of system
values:

$$\mathbf{R}(i,j) = \Theta\left(\epsilon - d(\mathbf{x}(t_i), \mathbf{x}(t_j))\right) \tag{34}$$

with $d(\cdot,\cdot)$ a distance or similarity metric (by convention, $d(\mathbf{x}(t_i), \mathbf{x}(t_i) = 0)$, $\Theta$ the Heaviside step function, and $\epsilon$ a similarity
threshold. If $d(\mathbf{x}(t_i), \mathbf{x}(t_j)) < \epsilon$, then $\mathbf{x}(t_j)$ and $\mathbf{x}(t_j)$ are similar and $\mathbf{R}(i,j) = 1$ represents a recurrence of the system. By
definition, $\mathbf{R}$ is symmetric and $\forall\ 1 \leq i \leq N$, $\mathbf{R}(i,i) = 1$. A conceptual example is shown on Figure 14-a.

Before calculating $\mathbf{R}$, the system $\mathbf{x}$ can first be reduced to a lower-dimensional space (as in section 4.2.2), by e.g., projecting $\mathbf{x}$
onto a set of principal components (Marwan et al., 2007; Mukhin et al., 2022) Note that $\mathbf{R}$ depends on the selected similarity
metric and on $\epsilon$. For a given system $\mathbf{x}$, $\mathbf{R}$ is thus not uniquely defined. Choosing a value for $\epsilon$ should be done carefully. $\epsilon$ should
be small enough to account for "real" recurrences in the data, but not to small to have enough recurrences to analyse. Several
rules of thumb have been proposed in the literature (see Marwan et al. (2007) and references therein). For instance, $\epsilon$ can be
selected by requiring an average recurrence rate of 1-10%. For weather and climate data, $\epsilon$ could also be selected based on
physical considerations and expert judgment.

RPs are a convenient way to visualise a system's trajectory, especially for multi-dimensional systems (like circulation fields).
RPs capture persistent behaviour: vertical lines indicate stationarity and diagonal lines (outside of the main diagonal) indicate
recurrence. RPs can thus highlight periods when the trajectory of a system roughly visits the same sequence of states or parts
of the phase space.

Several metrics have been proposed to quantify the presence of specific patterns in RPs. They are known collectively as
"recurrence quantification analysis" (Marwan, 2008). Common ones include the recurrence rate:

$$RR = \frac{1}{N^2} \sum_{1 \leq i,j \leq N} \mathbf{R}(i,j) \tag{35}$$

the determinism, which quantifies the fraction of recurrence points forming diagonal lines of minimum length $l_{min}$:

$$DET = \frac{\sum_{l \geq l_{min}} lD(l)}{\sum_{l \geq 1} lD(l)} \tag{36}$$

where $D(l)$ is the number of diagonal lines of length $l$, and the laminarity, which quantifies the fraction of recurrence points
forming vertical lines of minimum length $l_{min}$:

$$LAM = \frac{\sum_{l \geq l_{min}} lV(l)}{\sum_{l \geq 1} lV(l)} \tag{37}$$



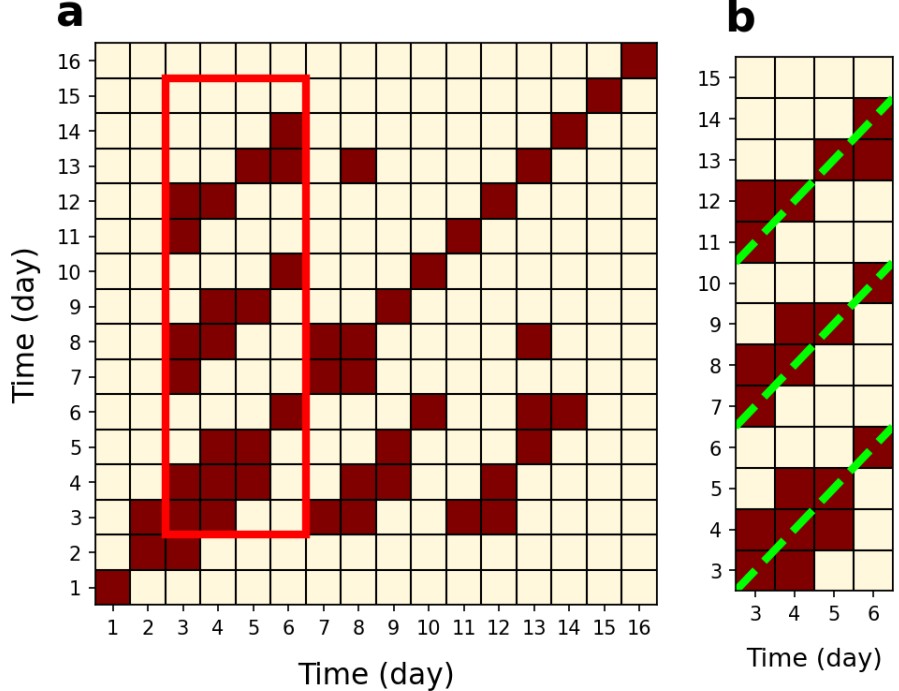

**Figure 14. Detecting recurrence with recurrence plots.** (a) Conceptual example of a recurrence plot. The red rectangle highlights a period during which the system exhibits recurrence with 3 successive similar 4-day sequences occurring over a 12-day period (highlighted in red). (b) Zoom over the recurrence period with the three separate 4-day sequences highlighted by green dashed lines.

where $V(l)$ is the number of vertical lines of length $l$. All three can be used as global measures of persistence (e.g., Ramirez-810 Amaro and Figueroa-Nazuno, 2006). However, these indices do not a priori discriminate between stationarity and persistence (high values of $RR$, for instance, could be due to either). To do so would require making sure that similar time steps are separated by "0"s in the RP (Figure 14-b).

RPs can also detect local recurrence in time. With RPs, one can extend the counting approach introduced in section 5.1 to multiple-day sequences of complex multi-dimensional data. Specifically, to search for sequences of $p$ steps that recur over 815 periods of $q$ steps ($0 < p < q$), we can simply define a recurrence index $RI_{p,q}(t)$ as the number of diagonal lines of length $p$ in the sub-matrix of $\mathbf{R}$ indexed by $[t, t+p] \times [t, t+q]$ (Figure 14-b). To make sure that distinct occurrences of the same sequence are separated, we can require diagonals to be separated by a minimum number of days (2 in Figure 14-b). It is also possible to allow for short breaks (e.g., 1 step) in the diagonals.

RPs have recently gained attention in a range of disciplines that deal with complex systems (Goswami, 2019), though only 820 rarely in environmental sciences. Ramirez-Amaro and Figueroa-Nazuno (2006) investigated the recurrence properties of major teleconnection patterns with RPs. Yiou et al. (2018) indirectly relied on RPs to identify the most recurrent states of North Atlantic circulation at the intra-seasonal timescale. Adeniji et al. (2018) used RPs to analyse the recurrence characteristics of



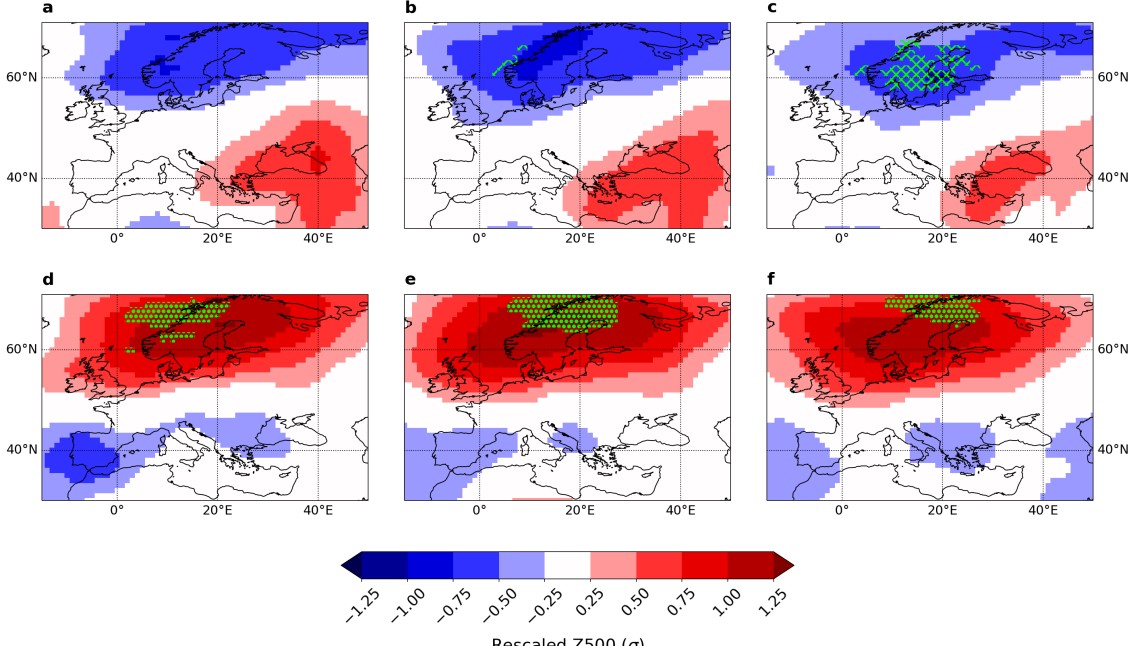

**Figure 15.** Two major 3-day recurring sequences of atmospheric circulation (Z500) over Europe during summer obtained by clustering the $\approx 30$ 21-day windows with $RI_{3,21}(t) \geq 2$ with the PAM algorithm. Similarity is measured with the SSIM index, with a similarity threshold $\epsilon$ in equation 34 of 0.25 (corresponding to a recurrence rate of $\approx 5\%$).

hourly wind speed in Nigeria and Ray et al. (2019) to investigate daily temperature and humidity data across India. Recently, Mukhin et al. (2022) used RPs to detect weather regimes in the Northern Hemisphere. On Figure 15, we show results obtained
for 3-day sequences of similar upper-level circulation patterns over Europe in summer. We calculate $RI_{p,q}(t)$ with $p = 3$ days and $q = 21$ days and identify the two most common sequences among the $\approx 30$ 21-day windows with $RI_{p,q}(t) \geq 2$. The method identifies recurring blocks and cyclones over Scandinavia as the most frequent recurring sequences of circulation patterns. The results appear physically meaningful since counts of individual blocks and cyclones during the corresponding windows range from 1-2 (for blocks) and 4-11 (for cyclones). More accurate patterns could possibly be obtained by choosing more than 2
clusters, but the results of Figure 15 are only meant to illustrate the potential of the method.



| Method | Section | Type | Definition | Features | Limitations |
|---|---|---|---|---|---|
| Window counts | 5.1 | State | Absolute event counts over windows of length $\tau$ | Simple, impact-driven approach; requires a fixed timescale | Basic characterisation of recurrence (e.g., no statistical significance) |
| Dispersion metrics | 5.2 | State | Characterise the degree of deviation in the distribution of event counts at a fixed timescale from homogeneous Poisson series | Statistical approach to recurrence | Not related to marginal event frequency; seasonality can make statistical significance assessment difficult |
| Ripley's K | 5.3 | State | Counts the average number of events in the neighbourhood of a given event | Statistical approach to recurrence; simple visualisation of recurrence across timescales; direct link to event counts (impacts) | Seasonality can make statistical significance assessment difficult |
| Inter-event times | 5.4 | State | Asymptotic characterisation of the distribution of inter-event times | Statistical approach to recurrence; discriminates short- and heavy tailed distributions where short-tailed ones are consistent with no-memory processes (recurrence occurs by chance) | Physical interpretation may be complex |
| Event count modeling | 5.5 | State | Fit a statistical model to $N(\tau)$ (event counts over windows of length $\tau$), e.g. Poisson model | Requires a fixed timescale $\tau$; allows to model the effects of covariates on recurrence | Requires statistical assumptions on the distribution of event counts |
| Recurrence plots | 5.6 | Global/ State/ Episodic | 2D binary matrix $\mathbf{R}$ describing system self-similarity at all time steps: $\mathbf{R}(i,j) = 1$ iff $d(\mathbf{x}(t_i),\mathbf{x}(t_j)) \leq \epsilon$; several global or local metrics can be defined | Requires a similarity threshold ($\epsilon$); very versatile approach that can be applied to any kind of data | Can be complex to implement and may require custom metrics depending on the application |

**Table 2.** As Table 1, but for recurrence methods.



**Summary and outlook**

Weather persistence on S2S timescales has been a topic of research since the early days of meteorology. Stationary or recurrent behaviour are common features of weather dynamics, and are strongly related to fundamental physical processes, weather predictability and surface weather impacts. Studying weather persistence is therefore important for theoretical as well as practical reasons. One challenge is that persistence remains a very broad concept that relates to different behaviours in dynamical systems. We propose a typology / structure for the broad concepts related to persistence. Namely that persistence is used to describe the average behaviour of a system, across its whole trajectory (global persistence); sometimes to refer to specific segments of this trajectory (episodic persistence); and sometimes to qualify the behaviour of particular system states (state persistence).

A wide range of methods have been introduced in the literature to describe persistence in weather and climate series, and several exist for each type of persistence. They offer many distinct and often complementary perspectives on persistence. Some methods quantify persistence in a statistical framework, which can be very relevant for weather forecasting. Others focus instead on persistent periods, including "statistical flukes", a useful approach for risk assessment but which says nothing about the overall behaviour of the series. Other methods yet aim at identifying which weather patterns tend to be more persistent than others.

The diversity of existing methods presented in this review reflects the fact that "persistence" is a multi-faceted concept. While we can agree on a general definition of the concept, many options exist when it comes to actually quantify persistence in real-world data. Though different methods may be related, each sheds light on a particular aspect of persistence. What is meant by "persistence" can not be dissociated from the metric used to quantify it. The choice of method should be guided by the end goal, whether process understanding, risk/impact assessment or predictability. Future research should nevertheless perhaps consider testing more systematically the robustness of results to the choice of persistence metric. This could matter particularly to better characterise potential trends in persistence under climate change and their associated impacts. As a final note, while we centred our review on S2S persistence, most of the methods we discussed apply in principle to other timescales, be they sub-daily or multi-decadal. What really differs is how to define the system to analyse (daily time steps may not be relevant for inter-annual variability, for instance) and how to interpret persistence.

*Code availability.* R code implementing several of the methods presented in this paper is available at https://github.com/Quriosity129/persistence. Code to calculate the dynamical system metrics (section 4.2.4) is available for R at https://github.com/thaos/dtheta or from the R package `CSTools` (Perez-Zanon and coauthors, 2022), and for Python at https://github.com/yrobink/CDSK. The Structural Similarity Index (SSIM) can be calculated with the Python package `scikit-image` (van der Walt et al., 2014).

**Appendix A: Data**

We illustrate some of the methods (Figures 2, 5, 7, 10 and 15) with data from the ERA5 reanalysis (Hersbach et al. (2020); available from https://dx.doi.org/10.24381/cds.bd0915c6). We use geopotential height at 500 hPa and 2-meter temperature





for the months of June, July and August, and over the 1979-2020 period. We remove their seasonality and long-term trends by normalising the data with a moving 30-day, 7-year window (as in Pfleiderer and Coumou (2018)). The calculation of the blocking and cyclone indices in Figure 15 is described in Tuel and Martius (2022a).

*Author contributions.* AT: Conceptualisation; Formal analysis; Visualisation; Software; Writing - original draft. OM: Conceptualisation; Funding acquisition; Supervision; Writing - review & editing.

*Competing interests.* O.M. serves as editor of Earth System Dynamics. The authors declare no further competing interests.

*Acknowledgements.* O.M. acknowledges support from the Swiss Science Foundation (SNSF) grant number 178751.



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
