# Peer review of "Weather persistence on sub-seasonal to seasonal timescales: a methodological review"

_EGUsphere, 2023_

## Referee Comment (RC1)

Reviewer Comment on: Weather persistence on sub-seasonal to seasonal timescales: a methodological review

Authors: Alexandre Tuel and Olivia Martius

Submitted to: egusphere-2023-111

**Summary**

The authors present a manuscript for a review article on persistence as the occurrence of approximately constant or recurrent atmospheric states in the subseasonal to seasonal (S2S) time range. Persistence determines damages, challenges understanding and promises predictability. The review considers various aspects of persistence: definitions, methodologies, and examples encompassed by this rather general notion.

In their review, the authors undertake a challenge when they combine meteorological observations and a mathematical definition in terms of advanced statistical and dynamical approaches. While a general definition of persistence is already hard, the related notion recurrence is even more difficult to grasp and to distinguish. Not surprisingly, the review is weak when the authors attempt the almost impossible task to provide a comprehensive theoretical framework, but it gets stronger when it resorts to methodologies and meteorological examples. In particular the variety of examples and their properties show the width of the seemingly simple notion persistence.

The authors are well aware of the difficulties involved and do not hide that. The review is worth to read for two reasons: the collection of methodologies that have been suggested to analyze persistence, and the huge amount of examples. In summary, the authors provide a broad and useful overview with a lot of insight. Below I mention some aspects that I noticed.

**Specific comments**

Line 55: Since persistence needs a timescale for the definition this question is somehow circular: 'At which timescale(s) does the persistence occur? There could be reference to Section 3.4.

Line 124, Eq $(\mathbf{x}(t))_t \in R^m$ what is the meaning of the subscript t together with the argument t? This should be explained.

Line 126, I recommend to use 'state space' instead of phase space (the often used notion phase space is reserved for Hamiltonian systems).

Line 136: To characterize the difference between precipitation and temperature an autocorrelation time would be appropriate here. This is better than 'inertia' since it is difficult to associate precipitation with an inertia, while temperature, on the other hand, has something like an inertia due to the heat capacity.

Lines 143, 155 in Section 3.1 *Global, state and episodic persistence.* It seems that global and state persistence are different concepts. Global persistence appears nothing else than stationarity which includes recurrence. Why is global persistence 'strongly related to intrinsic system predictability'? Maybe these definitions could be useful: Does persistence of a state x(t) mean dx(t)/dt=0. And could global persistence be defined in terms of integrals or averages like d<x(t)>/dt=0? And are conservation laws useful?

Line 184: What is a 'symmetrically, persistent state'?

Line 188: How can this sentence be understood: 'However, state persistence only characterizes the average behavior of system states.'

Line 201: Section 3.2 *Lagrangian and Eulerian perspectives.*

Here is a good opportunity to define Lagrangian stationarity of a quantity $\psi(\mathbf{x},t)$ along a flow $\mathbf{u}$, by $\partial\psi/\partial t + \mathbf{u}\cdot\nabla\psi = 0$, in comparison to the Eulerian stationarity with $\partial\psi/\partial t = 0$.

Line 216: The authors write 'self-similarity of system values $\mathbf{x}(t)$ with a metric' but this notion can be confused with geometric self-similarity used in the definition of fractal objects for example. Or is that what the authors intend?

Line 300: write what the symbol $E(\ldots)$ denotes.

Line 348: On Long-range memory: mention the absence of a timescale.

Line 360: Write that the Hurst exponent H and d are related by H=2d+1, see the Table 1 in Franzke et al. (2020).

Line 382: Written is: 'If temporal dependence is present'. This is unclear. Eq. (10) is the result a power-law in of $\rho$ (Eq.(6)), line 350.

Line 490: Section 4.2.4 *Extreme values and dynamical systems theory*. The role of the extremal index $\theta$ is difficult here. In extreme value statistics the extremal index is the inverse time scale with co-occurring extreme events, used to eliminate short term variability and to combine events. Why is this index a measure of stationarity? Is it correct that short term excursions are allowed with $\theta$ in Eq (13)?

Line 569: Please write that $R_{\mathbf{x}0}$ is the residence time for state $\mathbf{x}_0$.

---

## Author Comment (AC1)

**Response to reviewer comments**
* * *
**Comment 1.1** *In their review, the authors undertake a challenge when they combine meteorological observations and a mathematical definition in terms of advanced statistical and dynamical approaches. While a general definition of persistence is already hard, the related notion recurrence is even more difficult to grasp and to distinguish. Not surprisingly, the review is weak when the authors attempt the almost impossible task to provide a comprehensive theoretical framework, but it gets stronger when it resorts to methodologies and meteorological examples. In particular the variety of examples and their properties show the width of the seemingly simple notion persistence.*
*The authors are well aware of the difficulties involved and do not hide that. The review is worth to read for two reasons: the collection of methodologies that have been suggested to analyze persistence, and the huge amount of examples. In summary the authors provide a broad and useful overview with a lot of insight. Below I mention some aspects that I noticed.*

**Answer**: Thank you for your positive comments. It has indeed not been an easy task to collect and arrange the various methods and definitions that have been used to describe the extremely broad concept of persistence. The goal of sections 2 and 3 is to introduce, with as few mathematical formalities and assumptions as possible, the different meanings that have been given to persistence, and the questions one should ask themselves when using this word. We realise that this discussion has its limitations, and a bit more formalism (as you suggested in your Comment 1.6 below, for instance, would probably be welcome.
* * *
**Comment 1.2** *Line 55: Since persistence needs a timescale for the definition this question is somehow circular: "At which timescale(s) does the persistence occur?" There could be reference to Section 3.4.*

**Answer**: You are right that the current formulation sounds a bit circular, so we could reformulate such as "What are the persistent timescales in the data?" As you say, persistence requires a timescale to be defined, but this timescale is not necessarily known or define a priori.
* * *
**Comment 1.3** *Line 124, Eq $(\mathbf{x}(t))_t \in R^m$ what is the meaning of the subscript $t$ together with the argument $t$? This should be explained.*

**Answer**: Thanks for pointing this out. We are simply referring to the dynamical system here, and wanted to point out that it was indexed by time, but this notation is unnecessarily complicated. We will replace by $\mathbf{x}(t)$.
* * *
**Comment 1.4** *Line 126, I recommend to use 'state space' instead of phase space (the often used notion phase space is reserved for Hamiltonian systems).*

**Answer**: This is a very good suggestion which we will adopt in the revised version. Thanks!
* * *
**Comment 1.5** *Line 136: To characterize the difference between precipitation and temperature an autocorrelation time would be appropriate here. This is better than 'inertia' since it is*

*difficult to associate precipitation with an inertia, while temperature, on the other hand, has something like an inertia due to the heat capacity.*

**Answer**: This example is meant as an illustration of the concept of global persistence (before introducing it a few lines later). Autocorrelation would indeed be very relevant to characterise the difference between temperature and precipitation (and it is one of the common "global persistence" methods). The term "inertia" is used here in a more general way than in the context of the temperature/precipitation example, to highlight the fact that global persistence of a dynamical system describes how "fast" it tends to evolve, in other words it refers to its inertia.
* * *
**Comment 1.6** *Lines 143, 155 in Section 3.1 Global, state and episodic persistence. It seems that global and state persistence are different concepts. Global persistence appears nothing else than stationarity which includes recurrence. Why is global persistence 'strongly related to intrinsic system predictability'? Maybe these definitions could be useful: Does persistence of a state x(t) mean dx(t)/dt=0. And could global persistence be defined in terms of integrals or averages like $d < x(t) > /dt = 0$? And are conservation laws useful?*

**Answer**: Global and state persistence are indeed two different concepts. Global persistence relates to the stationarity/recurrence of a time series, but assessed over all the time steps of the series, while state persistence relates to the stationarity/recurrence of a specific system state. Your suggestion to illustrate these definitions with equations is very useful! In that sense, global persistence of a system $\mathbf{x}(t)$ looks at whether $\left\langle \frac{d\mathbf{x}(t)}{dt} \right\rangle = 0$ is, while the (state) persistence of system state $\mathbf{x}_0(t)$ looks at whether $\left\langle \frac{d\mathbf{x}(t)}{dt} \big|_{\mathbf{x}(t)=\mathbf{x}_0} \right\rangle = 0$ (with $<>$ being a time average). Whether conservation laws are useful in general is hard to say. It is quite possible that persistent system states or persistent periods are characterised by the (quasi)-conservation of some global quantity, but the answer really depends on the system under analysis.
* * *
**Comment 1.7** *Line 184: What is a 'symmetrically, persistent state'?*

**Answer**: This is a misunderstanding, "symmetrically" does not refer to the state but is used to highlight the connection to the previous sentence. To avoid this problem, we can replace "symmetrically" by "correspondingly".
* * *
**Comment 1.8** *Line 188: How can this sentence be understood: 'However, state persistence only characterizes the average behavior of system states.'*

**Answer**: What we mean by this sentence is that state persistence relates to the overall tendency of a given system state to be persistent, and does not imply that all occurrences of this state will necessarily be persistent. For instance, extreme warm conditions in London may on average last a week, which can be considered to be persistent, and in that case the state "very warm conditions" would be a persistent state of the time series of daily temperatures in London, but it may happen that extreme warm conditions last only one or two days. We will reformulate the manuscript to make this clearer.
* * *
**Comment 1.9** *Line 201: Section 3.2 Here is a good opportunity to define Lagrangian stationarity of a quantity $\psi(\mathbf{x}, t)$ along a flow $\mathbf{u}$, by $\partial\psi/\partial t + \mathbf{u} \cdot \nabla\psi = 0$, in comparison to the Eulerian stationarity with $\partial\psi/\partial t = 0$.*

**Answer**: Thanks for this excellent suggestion, which we will make sure to adopt in the revision.
* * *
**Comment 1.10** *Line 216: The authors write 'self-similarity of system values* $\mathbf{x}(t)$ *with a metric' but this notion can be confused with geometric self-similarity used in the definition of fractal objects for example. Or is that what the authors intend?*

**Answer**: We hadn't thought of that, but you are right, it could be confusing for some readers. What we mean by "self-similarity" is the tendency for successive values of a dynamical system to be similar to each other, similar being understood in the sense of "resemblance". It is quite difficult to find an alternative. The use of "similarity" is well-accepted in e.g. "similarity metrics" which quantify the similarity, or likeness, between two objects. However, we could add a note to make it clear that our use of the word is different from its acceptation in geometry.
* * *
**Comment 1.11** *Line 300: write what the symbol E(...) denotes.*

**Answer**: $\mathcal{E}$ denotes the expectancy with respect to the distribution of $X_t$. We will make sure to specify it in the revision.
* * *
**Comment 1.12** *Line 348: On Long-range memory: mention the absence of a timescale.*

**Answer**: We mention this on l.355 when discussing the limitations of this approach.
* * *
**Comment 1.13** *Line 360: Write that the Hurst exponent H and d are related by H=2d+1, see the Table 1 in Franzke et al. (2020).*

**Answer**: Thanks for the suggestion, we will adopt it in the revision.
* * *
**Comment 1.14** *Line 382: Written is: 'If temporal dependence is present'. This is unclear. Eq. (10) is the result a power-law in of $\rho$ (Eq.(6)), line 350.*

**Answer**: We are not sure what you mean by this comment. What we wanted to say is that for pure random noise, the spectral density is constant, while for series with non-zero autocorrelation (at least at small lags), the power spectrum will often (but not necessarily) exhibit a power-law dependence as a function of frequency.
* * *
**Comment 1.15** *Line 490: Section 4.2.4. The role of the extremal index $\theta$ is difficult here. In extreme value statistics the extremal index is the inverse time scale with co-occurring extreme events, used to eliminate short term variability and to combine events. Why is this index a measure of stationarity? Is it correct that short term excursions are allowed with $\theta$ in Eq (13)?*

**Answer**: You are correct that in extreme value statistics, $\theta$ is typically used to decluster extreme event occurrences in peak-over-threshold analysis. However, the use of the extremal index as a measure of temporal persistence has been introduced by Lucarini et al. (2016) and has been extensively used by the same authors in a large number of papers. The idea is that a high $\theta$ indicates that event occurrences tend to be isolated, while a low $\theta$ indicates that events occur as part of a larger group. Equation (13) relates to a marginal probability on $\mathbf{x}(t)$ – it does not explicitly describe the length of event clusters. However, in extreme value theory it can be shown

that $\theta$ defined as in Equation (13) is approximately equal to the inverse of the mean cluster size (e.g., Moloney et al. https://doi.org/10.1063/1.5079656. Short-term excursions are allowed in the sense that the relationship between $\theta$ and the mean cluster size is approximate.
* * *
**Comment 1.16** *Line 569: Please write that $R_{X_0}$ is the residence time for state $x_0$.*

**Answer**: Thanks for the suggestion, we will adopt it in the revision.

---

## Author Comment (AC2)

**Response to reviewer comments**
* * *
**Comment 2.1** *The authors have put great effort in an attempt to bring together all the different blocs of persistence. Persistence is a loose concept as there is no definite definition for it, and the authors propose to bring together its different facets. I had to go more than once to get a clear picture of the different bits and pieces. For example, in the 6 sections there are 13 subsections (excluding subsub-subsections). Perhaps to aid readers, especially early career researchers, an extra figure showing a tree-like diagram linking the different concepts of persistence would be welcome.*

**Answer**: Thank you for your comment. This is an excellent suggestion which we will include in a revised version. It will certainly be helpful to readers (i) to get a clear overview of the paper and (ii) to easily navigate through the maze of methods which we discuss.
* * *
**Comment 2.2** *Some related papers are missing from this review. Two particular references related to persistence and therefore predctability: predictive oscillation patterns (Kooperberg and O'Sullivan 1996), and a related paper: optimally interpolated patterns (Hannachi 2008), making use of the power spectra (ie autocorrelation.) In global stationarity, third-order statistics based, eg on bispectrum (Pires and Hannachi 2021) would complement the persistence description. In relation to extremes and persistence archetypal analysis Hannachi and Trendafilov 2017, Chapman et a. 2022) also identifies 'quasi-stationary' states or regimes.*

**Answer**: Thank you for pointing out to us these additional papers, which we will include in the revision.
* * *
**Comment 2.3** *Pg. 1, abstract: delete acronym S2S. It is abbreviated in section 1.*

**Answer**: If the abstract needs to stand on its own, then the acronym should remain. So far as we know, the rule is that if an acronym is used in the abstract, it must be defined in the abstract, and then defined again the first time it is used in the main text.
* * *
**Comment 2.4** *Pg 3, l75: you mean Fig. 1c.*

**Answer**: Correct, thanks for noticing.
* * *
**Comment 2.5** *Pg7, l175: delete 'of'*

**Answer**: Thanks, will do.
* * *
**Comment 2.6** *Pg 13, eq (15): may be use $0 < |\alpha| < 1$ to include the case of anti-persistence.*

**Answer**: Good suggestion, thanks.
* * *
**Comment 2.7** *l358: add the following reference on short timescale of precipitation (Hannachi 2014).*

**Answer**: Thanks for pointing out this reference to us.

**Comment 2.8** *l367, 369: may be 'persistence' is more convenient here than 'stationary'.*

**Answer**: Good point. We will change to "persistent".
* * *
**Comment 2.9** *Pg 14, last paragraph 4.1.2: Fig.3 does not seem to have been mentioned.*

**Answer**: This is an oversight; we will add the reference to Fig. 3 at l. 389 when mentioning the Fraedrich and Larnder study.
* * *
**Comment 2.10** *Pg 15, l407: consider adding an earlier reference of Baur (1951) on Grosswetterlagen.*

**Answer**: Good suggestion, we will add this reference to the sentence.
* * *
**Comment 2.11** *Pg18, l461: add "by projecting simplified dynamics (eg quasi-geostrophy) onto the leading modes of variability of the GCM simulation" after "January conditions".*

**Answer**: Thanks for this detauil, we will add it to the revision.
* * *
**Comment 2.12** *Pg 20, eq (13): x is not specified, and I think there is some confusion in the original reference (Faranda et al.) I think x represents the log of the distance between state at time $t_{\mathbf{x(t)}}$ and $t_{\mathbf{x_0}}$.*

**Answer**: You are right (and there is indeed a confusion in the original reference). The distribution is fitted to the distance values, and we will modify the corresponding equation as follows:

$$\mathbb{P}\left(d(\mathbf{x}(t), \mathbf{x}_0) \leq \epsilon\right) \simeq \exp\left\{-\theta(\mathbf{x}_0)\frac{d(\mathbf{x}(t), \mathbf{x}_0) - \mu(\mathbf{x}_0)}{\sigma(\mathbf{x}_0)}\right\} \tag{1}$$
* * *
**Comment 2.13** *Pg 23, eq(21): my understanding is that $\alpha$ is a probability $(P(x_0) = P(x_{t+1} = x_0 | x_t = x_0))$, therefore eq(21) < 0, and also $\log(\alpha) < 0$, please clarify.*

**Answer**: You are right, we made a mistake in this equation... We have

$$\mathbb{P}\left(R_{\mathbf{x}_0} \geq n\right) = \beta \times \alpha^{n-1}$$

and therefore

$$\mathbb{P}\left(R_{\mathbf{x}_0} = n\right) = \mathbb{P}\left(R_{\mathbf{x}_0} \geq n\right) - \mathbb{P}\left(R_{\mathbf{x}_0} \geq n+1\right) = \beta\alpha^{n-1}(1-\alpha) > 0$$

The slope is indeed negative with respect to $n$ since the probability of the residence time being equal to $n$ decreases with $n$.
* * *
**Comment 2.14** *Pg 31, l646: Ripley's K function was considered in Stephenson et al. (2004), and Hannachi (2010).*

**Answer**: Thanks for pointing out these two references, we will add the following sentence to the section: "*Stephenson et al. (2004) and Hannachi (2010) also used Ripley's K to characterise the clustering of climate modes in the phase space.*"
* * *
**Comment 2.15** *Pg 31, l686: " ... as the variance of successive event counts over an interval of ..."*

**Answer**: We will correct it, thanks.
* * *
**Comment 2.16** *Pg 37, l791: full stop before 'Note'*

**Answer**: Will correct, thanks.
* * *
**Comment 2.17** *l793: 'too'*

**Answer**: Will correct, thanks.
* * *
**Comment 2.18** *l801: use 'measures' instead of 'metrics'*

**Answer**: Will correct, thanks.
* * *
**Comment 2.19** *Fig. 15: are a, b, c 1-day apart?*

**Answer**: yes, indeed. We will make sure to make it explicit in the figure caption.
* * *
**Comment 2.20** *Section 5: I suggest renumbering/relabelling the subsections as:*

*5.1 Diagnostic methods*
 *5.1.1 Window counts*
 *5.1.2 Dispersion metrics*
 *5.1.3 Ripley's K function*
 *5.1.4 Distribution of inter-event times*
*5.2 Stochastic modeling of recurrence*
 *5.2.1 Events series*
 *5.2.2 Recurrence plots*

**Answer**: This is an excellent suggestion which brings more structure to this section. Thanks!

---

## Author Response (AR1)

**Response to comments by reviewer #1**
* * *
**Comment 1.1** *Line 55: Since persistence needs a timescale for the definition this question is somehow circular: "At which timescale(s) does the persistence occur?" There could be reference to Section 3.4.*

**Answer**: We reformulate this sentence as "What are the persistent timescales in the data?"
* * *
**Comment 1.2** *Line 124, Eq $(\mathbf{x}(t))_t \in R^m$ what is the meaning of the subscript $t$ together with the argument $t$? This should be explained.*

**Answer**: Thanks for pointing this out. We are simply referring to the dynamical system here, and wanted to point out that it was indexed by time, but this notation is unnecessarily complicated. We replaced by $\mathbf{x}(t)$.
* * *
**Comment 1.3** *Line 126, I recommend to use 'state space' instead of phase space (the often used notion phase space is reserved for Hamiltonian systems).*

**Answer**: We adopted this suggestion throughout the revision, thanks.
* * *
**Comment 1.4** *Line 136: To characterize the difference between precipitation and temperature an autocorrelation time would be appropriate here. This is better than "inertia" since it is difficult to associate precipitation with an inertia, while temperature, on the other hand, has something like an inertia due to the heat capacity.*

**Answer**: This example is meant as an illustration of the concept of global persistence (before introducing it a few lines later). Autocorrelation would indeed be very relevant to characterise the difference between temperature and precipitation (and it is one of the common "global persistence" methods). The term "inertia" is used here in a more general way than in the context of the temperature/precipitation example, to highlight the fact that global persistence of a dynamical system describes how "fast" it tends to evolve, in other words it refers to its inertia.
* * *
**Comment 1.5** *Lines 143, 155 in Section 3.1 Global, state and episodic persistence. It seems that global and state persistence are different concepts. Global persistence appears nothing else than stationarity which includes recurrence. Why is global persistence "strongly related to intrinsic system predictability"? Maybe these definitions could be useful: Does persistence of a state $x(t)$ mean $dx(t)/dt = 0$. And could global persistence be defined in terms of integrals or averages like $d < x(t) > /dt = 0$? And are conservation laws useful?*

**Answer**: Global and state persistence are indeed two different concepts. Global persistence refers to the average behaviour of the system, while state persistence refers to the behaviour of specific states of the system. Global persistence translates either into very little change from one time step to the next (for stationarity) or in a tendency to the recurrence of system states across the trajectory. Global persistence is thus strongly related to system predictability since current values of the system strongly constrain its future values.

In mathematical terms, global stationarity doesn't necessarily translate into $dx(t)/dt \approx 0$, but instead into $\left\langle \left| \frac{d\mathbf{x}(t)}{dt} \right| \right\rangle \ll \frac{\sigma_x}{T}$ where $< \cdot >$ is a time average, $\sigma_x$ is the standard deviation of the series $x(t)$ and $T$ its typical timescale of evolution. This means that variability of the system at small timescales $((x(t + dt) - x(t))/dt)$ is small compared to its variability at long timescales. State persistence can simply be understood as the tendency for the system to stagnate in certain parts of the phase space. Whether conservation laws are useful in general is hard to say. It is quite possible that persistent system states or persistent periods are characterised by the (quasi)-conservation of some global quantity, but the answer really depends on the system under analysis. We slightly expanded the corresponding section by adding "*Global stationarity translates into the tendency for the system to change little at small timescales (successive values being close to each other). In mathematical terms, this translates into $\left\langle \left| \frac{d\mathbf{x}(t)}{dt} \right| \right\rangle \ll \frac{\sigma_x}{T}$ where $< \cdot >$ is a time average, $\sigma_x$ is the standard deviation of the series $x(t)$ and $T$ its typical timescale of evolution.*", and further: "*Global persistence is, however, strongly related to intrinsic system predictability, since present values of the system largely constrain its future values.*"
* * *
**Comment 1.6** *Line 184: What is a 'symmetrically, persistent state'?*

**Answer**: This is a misunderstanding, "symmetrically" does not refer to the state but is used to highlight the connection to the previous sentence. We replaced "symmetrically" with "correspondingly".
* * *
**Comment 1.7** *Line 188: How can this sentence be understood: 'However, state persistence only characterizes the average behavior of system states.'*

**Answer**: Following our initial reply, we reformulated the sentence as follows: "*However, state persistence only characterises the average behaviour of system states – it does not imply that all occurrences of a persistent state will necessarily be persistent.*"
* * *
**Comment 1.8** *Line 201: Section 3.2 Here is a good opportunity to define Lagrangian stationarity of a quantity $\psi(\mathbf{x}, t)$ along a flow $\mathbf{u}$, by $\partial\psi/\partial t + \mathbf{u} \cdot \nabla\psi = 0$, in comparison to the Eulerian stationarity with $\partial\psi/\partial t = 0$.*

**Answer**: Thanks for this excellent suggestion. We added the following sentence in the revision: "*The Eulerian stationarity of a quantity $\psi(\mathbf{x}, t)$ translates into $\partial\psi/\partial t = 0$ while the Lagrangian stationarity implies $\partial\psi/\partial t + \mathbf{u} \cdot \nabla\psi = 0$ where $\mathbf{u}$ is the background flow.*"
* * *
**Comment 1.9** *Line 216: The authors write "self-similarity of system values $\mathbf{x}(t)$ with a metric" but this notion can be confused with geometric self-similarity used in the definition of fractal objects for example. Or is that what the authors intend?*

**Answer**: Following our earlier reply, we added the following to avoid any confusion: "*By self-similarity, we refer to the tendency for successive values of $\mathbf{x}(t)$ to be similar to each other, according to some metric. This is not to be confused with the concept of geometric self-similarity in fractal geometry.*"
* * *
**Comment 1.10** *Line 300: write what the symbol $E(\dots)$ denotes.*

**Answer**: We added "$\mathcal{E}$ *denotes the expectancy with respect to the distribution of $X_t$*."
* * *
**Comment 1.11** *Line 348: On Long-range memory: mention the absence of a timescale.*

**Answer**: We mention this at l.355 when discussing the limitations of this approach.
* * *
**Comment 1.12** *Line 360: Write that the Hurst exponent H and d are related by H=2d+1, see the Table 1 in Franzke et al. (2020).*

**Answer**: We modified the last sentence of the paragraph as follows: "*H is also theoretically related to the dependency parameter (as $H = 2d + 1$) and the power spectrum exponent (see below) (Franzke et al., 2020)*".
* * *
**Comment 1.13** *Line 382: Written is: "If temporal dependence is present". This is unclear. Eq. (10) is the result a power-law in of $\rho$ (Eq.(6)), line 350.*

**Answer**: We are not sure what you mean by this comment. What we wanted to say is that for pure random noise, the spectral density is constant, while for series with non-zero autocorrelation (at least at small lags), the power spectrum will often (but not necessarily) exhibit a power-law dependence as a function of frequency.
* * *
**Comment 1.14** *Line 490: Section 4.2.4. The role of the extremal index $\theta$ is difficult here. In extreme value statistics the extremal index is the inverse time scale with co-occurring extreme events, used to eliminate short term variability and to combine events. Why is this index a measure of stationarity? Is it correct that short term excursions are allowed with $\theta$ in Eq (13)?*

**Answer**: Building on our initial reply, we added the following sentence to the paragraph to make the link bewteeen extreme value statistics and the dynamical systems approach: "*In extreme value statistics, $\theta$ is the inverse average duration of consecutive sequences of extreme events, and is used to cluster events (Ferro and Segers, 2003). A large $\theta$ therefore indicates that event occurrences tend to be isolated, while a low $\theta$ indicates that events occur as part of a larger group.*"
* * *
**Comment 1.15** *Line 569: Please write that $R_{X_0}$ is the residence time for state $x_0$.*

**Answer**: Thanks, corrected.

**Response to comments by reviewer #2**
* * *
**Comment 2.1** *The authors have put great effort in an attempt to bring together all the different blocs of persistence. Persistence is a loose concept as there is no definite definition for it, and the authors propose to bring together its different facets. I had to go more than once to get a clear picture of the different bits and pieces. For example, in the 6 sections there are 13 subsections (excluding subsub-subsections). Perhaps to aid readers, especially early career researchers, an extra figure showing a tree-like diagram linking the different concepts of persistence would be welcome.*

**Answer**: Thank you for your comment. This is an excellent suggestion which we included in the revised version (see Figure R1 in this reply).

[Figure]

Figure R1: Overview of the persistence methods discussed in this paper. Section numbers relative to each methods are indicated in bold between brackets.
* * *
**Comment 2.2** *Some related papers are missing from this review. Two particular references related to persistence and therefore predictability: predictive oscillation patterns (Kooperberg and O'Sullivan 1996), and a related paper: optimally interpolated patterns (Hannachi 2008), making use of the power spectra (i.e. autocorrelation.) In global stationarity, third-order statistics based, e.g. on bispectrum (Pires and Hannachi 2021) would complement the persistence description. In relation to extremes and persistence archetypal analysis (Hannachi and Trendafilov 2017, Chapman et al. 2022) also identifies 'quasi-stationary' states or regimes.*

**Answer**: Thank you for pointing out to us these additional papers, which we included in the revision. We added a reference to Predictive Oscillation Patterns and Optimally Interpolated Patterns in section 4.2.3., and a reference to archetypal analysis in section 4.2.1. We also

mentioned the bicorrelation and bispectrum in the global stationarity section (4.1.1 and last paragraph of 4.1.2).
* * *
**Comment 2.3** *Pg. 1, abstract: delete acronym S2S. It is abbreviated in section 1.*

**Answer**: If the abstract needs to stand on its own, then the acronym should remain. So far as we know, the rule is that if an acronym is used in the abstract, it must be defined in the abstract, and then defined again the first time it is used in the main text.
* * *
**Comment 2.4** *Pg 3, l75: you mean Fig. 1c.*

**Answer**: Thanks, corrected.
* * *
**Comment 2.5** *Pg7, l175: delete 'of'*

**Answer**: Thanks, corrected.
* * *
**Comment 2.6** *Pg 13, eq (15): may be use $0 < |\alpha| < 1$ to include the case of anti-persistence.*

**Answer**: Thanks, adopted.
* * *
**Comment 2.7** *l358: add the following reference on short timescale of precipitation (Hannachi 2014).*

**Answer**: Reference added, thanks.
* * *
**Comment 2.8** *l367, 369: may be 'persistence' is more convenient here than 'stationary'.*

**Answer**: We changed to "persistent".
* * *
**Comment 2.9** *Pg 14, last paragraph 4.1.2: Fig.3 does not seem to have been mentioned.*

**Answer**: We added the reference to Fig. 3 at l. 389 when mentioning the Fraedrich and Larnder study.
* * *
**Comment 2.10** *Pg 15, l407: consider adding an earlier reference of Baur (1951) on Grosswetterlagen.*

**Answer**: Reference added, thanks.
* * *
**Comment 2.11** *Pg18, l461: add "by projecting simplified dynamics (eg quasi-geostrophy) onto the leading modes of variability of the GCM simulation" after "January conditions".*

**Answer**: We modified the sentence as follows: "*Haines and Hannachi (1995) and Hannachi (1997) estimated quasi-stationary states over the North Pacific in the output from a global climate forced by perpetual January conditions, by projecting simplified dynamics (e.g., quasi-geostrophy) onto the leading modes of variability of the GCM simulation.*"
* * *
**Comment 2.12** *Pg 20, eq (13): x is not specified, and I think there is some confusion in the original reference (Faranda et al.) I think x represents the log of the distance between state at time $t_{\mathbf{x(t)}}$ and $t_{\mathbf{x_0}}$.*

**Answer**: You are right (and there is indeed a confusion in the original reference). The distribution is fitted to the distance values, and we modified the corresponding equation as follows:

$$\mathbb{P}\left(d(\mathbf{x}(t), \mathbf{x}_0) \leq \epsilon\right) \simeq \exp\left\{-\theta(\mathbf{x}_0)\frac{d(\mathbf{x}(t), \mathbf{x}_0) - \mu(\mathbf{x}_0)}{\sigma(\mathbf{x}_0)}\right\} \tag{1}$$
* * *
**Comment 2.13** *Pg 23, eq(21): my understanding is that $\alpha$ is a probability $(P(x_0) = P(x_{t+1} = x_0 | x_t = x_0))$, therefore eq(21) < 0, and also $\log(\alpha) < 0$, please clarify.*

**Answer**: You are right, we made a mistake in this equation... We have

$$\mathbb{P}\left(R_{\mathbf{x}_0} \geq n\right) = \beta \times \alpha^{n-1}$$

and therefore

$$\mathbb{P}\left(R_{\mathbf{x}_0} = n\right) = \mathbb{P}\left(R_{\mathbf{x}_0} \geq n\right) - \mathbb{P}\left(R_{\mathbf{x}_0} \geq n+1\right) = \beta\alpha^{n-1}(1 - \alpha) > 0$$

The slope is indeed negative with respect to $n$ since the probability of the residence time being equal to $n$ decreases with $n$.
* * *
**Comment 2.14** *Pg 31, l646: Ripley's K function was considered in Stephenson et al. (2004), and Hannachi (2010).*

**Answer**: Thanks for pointing out these two references, we added the following sentence to the section: "*Stephenson et al. (2004) and Hannachi (2010) also used Ripley's K to characterise the clustering of climate modes in the state space.*"
* * *
**Comment 2.15** *Pg 31, l686: " ... as the variance of successive event counts over an interval of ..."*

**Answer**: We modified the sentence as follows: "*Similarly, the Allan Factor (AF) is defined as the variance of successive event counts over an interval of length $\tau$ divided by twice the average event count in $\tau$ steps*".
* * *
**Comment 2.16** *Pg 37, l791: full stop before 'Note'*

**Answer**: Corrected, thanks.
* * *
**Comment 2.17** *l793: 'too'*

**Answer**: Corrected, thanks.
* * *
**Comment 2.18** *l801: use 'measures' instead of 'metrics'*

**Answer**: Corrected, thanks.
* * *
**Comment 2.19** *Fig. 15: are a, b, c 1-day apart?*

**Answer**: Correct, we specified it in the caption (*"Note that successive panels (a, b, c, and d, e, f) are 1-day apart."*).
* * *
**Comment 2.20** *Section 5: I suggest renumbering/relabelling the subsections as:*

*5.1 Diagnostic methods*
    *5.1.1 Window counts*
    *5.1.2 Dispersion metrics*
    *5.1.3 Ripley's K function*
    *5.1.4 Distribution of inter-event times*
*5.2 Stochastic modeling of recurrence*
    *5.2.1 Events series*
    *5.2.2 Recurrence plots*

**Answer**: This is an excellent suggestion which we adopted in the revision.

---

## Author Response (AR2)

**Answer to reviewers**

**Manuscript egusphere-2023-111**
**submitted to**
**Earth System Dynamics**

July 13, 2023

Dear Dr. Franzke,

Please find attached our last responses to reviewer comments and our revised manuscript. We hope this revised version will address all remaining concerns.

We take this opportunity to thank you and the reviewers for your time and constructive comments, which considerably improved our manuscript.

Best wishes,

Alexandre Tuel and Olivia Martius

**Response to comments by reviewer #1**

**Comment 1.1** *I think the paper is suitable for publications as is. There is a small misprint. The section number 5.1.1 is repeated twice. The second one should be numbered 5.1.2*

**Answer**: Thank your for your useful comments during this review process. The typo is now fixed.

**Response to comments by reviewer #2**
* * *
**Comment 2.1** *The added first figure did make the overall concept clearer. However, the author did not reference the figure in the main text. A related issue is whether the author could create another figure or modify Figure 1 to include all the sections, rather than just Section 4 and Section 5.*

**Answer**: Thank you for this comment. We added a reference to this figure at the end of the introduction. We also updated the figure to refer to section 2 as well. However, for the sake of keeping it legible, we would rather not make it more complex.
* * *
**Comment 2.2** *Regarding the indication of equations, there seems to be inconsistency in the author's approach. Sometimes the authors simply mention the equation number in the sentence, while other times they do not. To reduce the chance of misunderstanding, I suggest that the author maintain consistency when referring to specific equations. For example, they could use the format "Equation 1," "Equation 2," or another form such as "Eq1," "Eq2."*

**Answer**: Good point. All equations are now referenced by "equation (n)".
* * *
**Comment 2.3** *In relation to the research questions (L54-57), it may be beneficial to discuss or provide clarification regarding stationarity or recurrence in this section.*

**Answer**: We expanded the introduction of the stationarity/recurrence topics at the beginning: "*Surface weather persistence at sub-seasonal to seasonal (S2S) timescales can have severe impacts on human and natural systems. Long-lasting dry conditions, for instance, can lead to droughts and wildfires and can affect agriculture and energy production. Long-lasting wet spells may cause severe flooding and crop loss. Persistent surface weather can result either from quasi-stationary, long-lived atmospheric circulation conditions (stationarity) or from repeated, shorter-lived circulation features (recurrence). Recurrence refers to the repeated occurrence of similar large-scale circulation types or weather systems within some (S2S) time interval, usually with brief interruptions. Many recent high-impact weather and climate events were linked to persistent quasi-stationary or recurrent weather conditions. An example for recurrence are the Western European floods in July 2021 that occurred at the end of an extreme wet spell in Western Europe. The wet spell resulted from repeated atmospheric blocks and Rossby wave breaking episodes (Tuel et al., 2022b). Other examples of recurrence include the floods in the UK during winter 2013-2014 and in Queensland (Australia) in February-April 2022 both were caused by sequences of cyclones (Huntingford et al., 2014) (Wikipedia, 2022; Floodlist, 2022). An example for quasi-stationarity is the catastrophic flooding in Pakistan in summer 2022 that resulted (in part) from long-lasting and particularly heavy monsoon rains (Mallapaty, 2022). Intense heatwaves and associated atmospheric circulations also tend to be persistent (often arising from a combination of recurrence and quasi-stationarity) (Lorenz et al., 2010), as in Western Europe in 2003 (Black et al., 2004; García-Herrera et al., 2010), Western Russia in 2010 (Drouard and Woollings, 2018; Di Capua et al., 2021) or China (WMO, 2022) and India (Bloomberg, 2022) in 2022.*"
* * *
**Comment 2.4** *L24: merge the citations.*

**Answer**: Done, thanks.

***Comment 2.5*** *L73: in spring 2019? – it seems you indicated 1979.*

**Answer**: It is indeed 1979, thanks for noticing this typo.
* * *
***Comment 2.6*** *L100-120: I suggest the author rephrase or emphasize main points at the beginning. It takes time to grasp the idea (could be my problem).*

**Answer**: Please see our answer to your Comment 2.3.
* * *
***Comment 2.7*** *L233: is the ()t consistent with the previous one?*

**Answer**: Since we specify "the set of values" then the index is not really needed. We changed to "*They require the set of all* $\mathbf{x}(t)$ *values...*"
* * *
***Comment 2.8*** *Figure 12: the caption: 1975-2004.*

**Answer**: Thanks, change made.
* * *
***Comment 2.9*** *Figure 16: what are the differences between the different shadings?*

**Answer**: Thanks for noticing. The stippling/hatching refer to positive blocking/cyclone frequency anomalies. We added the following sentence to the caption: "*Stippling (respectively hatching) indicates atmospheric blocking (respectively cyclone) frequency anomalies larger than 30%.*".